# The evolution and convergence of mutation spectra across mammals
Andrea Talenti ✉, Toby Wilkinson, Liam J. Morrison & James G. D. Prendergast ✉

Despite the key role genetic mutations play in shaping phenotypic differences between species, little is currently known about the evolution of germline mutation spectra across mammals. Domesticated species are likely particularly interesting case studies because of their high mutation rates and complex evolutionary histories, which can span multiple founding events and genetic bottlenecks. Here we have developed a new reusable workflow, nSPECTRa, that can undertake the key steps in characterising mutation spectra, from determining ancestral alleles to characterising multiple forms of variation. We apply nSPECTRa to seven species, including several that have undergone domestication, and highlight how nSPECTRa can provide important insights into mutation rate evolution. While mutation spectra most often show marked differences between species and even breeds, certain mutation types have risen to a high frequency in subpopulations of different species, indicative of convergent evolution in mutation rates. This includes the previously characterized TCC-> TTC change enriched among European humans, which is also enriched among East Asian cattle. We show Indicine cattle are particularly interesting examples of how different mutation spectra segregate within a population and subsequently spread across the globe. Together, this work has important implications for understanding the control and evolution of mammalian mutation rates.

As a key driver of phenotypic differences between individuals, and the main substrate upon which selection acts, characterising the processes that generate germline mutations is vital to determine the processes driving traits, diseases and species evolution. Owing to the importance of DNA fidelity in mammals, there are several hundred genes[1] involved in DNA repair, spanning at least seven different repair pathways[2], from mismatch repair to non-homologous end joining. Each of these different pathways is preferentially associated with the repair of a different spectrum of DNA changes, and because of the large number of genes and pathways involved, natural genetic variation across them is expected to lead to differences in the efficiency with which different mutation types are repaired between individuals. Although genetic variants with a large effect on DNA repair have been observed, most notably within families with high incidences of cancer[3], these are generally rare in any given population, with the majority of genetic polymorphisms affecting DNA repair between individuals likely to have a small effect[4]. Nevertheless, the accumulation of multiple mutations across hundreds of repair genes over time could potentially lead to noticeable differences in the spectra of the mutations found between individuals. Supporting this idea, Harris and Pritchard[5] found that different human populations preferentially carry different numbers of DNA mutations in different K-mer contexts. For example, human European populations have

been observed to carry a relatively greater proportion of TCC > TTC changes, suggesting that individuals from this population have less efficiently corrected such changes over recent human history.

In addition to varying rates by sequence context, the distribution of single-nucleotide variants (SNVs) has been shown to be uneven across the genome, with directly adjacent SNVs occurring more often than expected and most often on the same DNA strand[6]. Although many of these neighbouring changes appear to be the result of a single mutational event in a single generation, that have been termed multi-nucleotide polymorphisms (MNPs)[7], more commonly adjacent SNVs are found at different frequencies in the population, suggesting that they are the result of two mutation events at different times leading to what has been termed sequential dinucleotide polymorphisms (SDMs)[6]. Consistent with this, flanking heterozygosity has been shown to be associated with an increased probability of base substitutions in the human genome[8], suggesting DNA mutations are often not independent of those that have previously occurred nearby. In previous work, these SDMs were shown to be even more differentiated between populations than single-nucleotide variants (SNVs) and to be driven via different mutational processes[6].

Beyond heritable variation in the ability to repair DNA damage, other factors also play a role in shaping the distribution of base changes in the

The Roslin Institute, Royal (Dick) School of Veterinary Studies, University of Edinburgh, Easter Bush Campus, Midlothian, UK. ✉e-mail: atalenti@ed.ac.uk; james.prendergast@roslin.ed.ac.uk

genome. Perhaps the most notable differences are in the rate at which certain sequences are prone to mutate[9]. Likewise, different environmental exposures to mutagens[10], natural and artificial selection, and biased gene conversion[11]. One of the largest drivers of DNA changes in mammals is the readiness with which methylated CpG sites deaminate to TpG dinucleotides, with such changes making up around a sixth of all observed germline mutations in humans[12,13]. Similarly, selection is expected to rapidly remove novel deleterious mutations from the population, potentially skewing the mutational profile of the remaining changes. However, as most novel mutations are expected to be neutral due to falling in non-coding, intergenic, or intronic regions, the impact of selection may not substantially skew the mutation profiles between individuals. This is especially true, as the location and number of functional regions are expected to be largely consistent across individuals, at least within species. Biased gene conversion (BGC), the process by which one copy of an allele is preferentially replaced by another through recombination[11], can also alter the spectrum of SNVs in the genome. This effect is known to be skewed towards increasing the frequency of C:G base pairs at the expense of A:T base pairs. However, recent studies have indicated that BGC is not likely to be a major driver of the observed differences in mutation spectra observed between individuals[14].

Although most studies have been carried out within species, by comparing the genomes of offspring to those of their parents, Bergeron et al.[15] estimated germline mutation rates across 68 vertebrate species. Factors such as generation time, age at sexual maturity, and other fecundity traits were observed to be linked to differences in the number of germline mutations per generation between species. Domesticated species were shown to be particularly interesting models for studying germline mutation profiles owing to their unusual evolutionary histories. The authors observed them to have unusually high mutation rates, which they attributed to a reduction in generation time in these species rather than any inherent differences in the underlying mutational processes. However, the relatively few mutations identified when comparing parental and offspring genomes means that it is not possible to compare the spectra of mutations in different K-mer contexts between species and populations using such trio-based approaches[15]. Furthermore, the non-model organisms often rely on lower-quality data, reducing the ability of bioinformatic tools to discriminate between relevant biological events and artefacts. The impact on mutation spectra of phenomena such as multiple domestication events in cattle[16], selection for phenotypic diversity in dogs[17], and population bottlenecks across domesticated species[18] is largely unknown.

Consequently, despite the importance of better understanding how patterns of mutation differ between individuals, more work is needed to study mutation-spectra differences outside of the major model organisms[14,15,19,20]. To date, this has been hampered due to a lack of suitable large, high-quality, whole-genome sequenced cohorts and easy-to-use software tools. However, factors such as the ever-decreasing costs of whole-genome sequencing and the availability of tools for studying SNV mutation rates, such as mutyper[21] and Relate[22], have now enabled the potential application of these approaches across a wider range of species. In this study, we expand on these tools to develop a unified, reusable workflow and analysis approach that can characterise different types of SNV and SDM mutation spectra in any diploid species and apply it to study the evolution and convergence of mutation spectra. We applied this workflow across several mammalian species, with a particular focus on domesticated animals, to characterise how mutation spectra have evolved and converged both within and across species.

## Results

### A reusable workflow for characterising mutation spectra across species

To enable the characterisation of mutation rate spectra across any diploid species, we generated a publicly available, reusable Nextflow workflow, nSPECTRa[23] (www.github.com/evotools/nSPECTRa). nSPECTRa enables users to run a range of analyses, including imputation, annotation, and calculation of both SNV and SDM spectra, and their relative rates over time.

Importantly, nSPECTRa can infer ancestral alleles for any species, a prerequisite for determining the direction of mutations, but which are generally unavailable for most species. The entire nSPECTRa workflow is shown in Fig. 1.

To demonstrate the utility of this pipeline and to explore the patterns of germline variation across mammals, we applied nSPECTRa to whole-genome variant calls of unrelated samples from seven different species (309 cattle, 175 African buffalo, 79 water buffalo, 36 horses, 350 pigs, 606 dogs, and 2561 humans). These species span a diverse range of mammals and cover three Bovidae and five domesticated species, enabling comparisons both within and across groups.

A key challenge when attempting to compare mutation spectra within and across species is that both the total number of mutations and the frequency of ancestral K-mers in the respective genomes differ between animals. This means that both factors need to be controlled for to enable comparisons of the relative rate at which different K-mers mutate across species. To achieve this, we adapted the median of ratios method originally developed for RNA-seq data analysis. Using this approach, the proportion of each K-mer in the genome showing a particular base change was first normalised by the occurrence of the corresponding K-mer in the ancestral genome. These normalised K-mer-specific mutation rates were further corrected by calculating a sample-wise median of ratios normalisation factor to control for differences in total mutation number between animals, as described by Anders and Huber[24]. As shown in Supplementary Fig. 1, this approach for processing mutation rates successfully places the mutation spectra of animals onto the same scale, so that they can be directly compared both within and across species.

### Divergence of SNV mutation profiles and impact on protein evolution across mammals

Figure 2a shows a Principal Component Analysis (PCA) representation of the relationship between the SNV mutation spectra across the seven studied species. All species showed a clear separation based on their spectra of germline SNV variation, highlighting the substantial divergence in mutational profiles across mammals. Consistent with their comparatively close evolutionary distances, the two domesticated and one wild Bovidae (cattle, water buffalo, and African buffalo) showed the most similar mutational profiles. Figure 2b shows the large differences between species across a diverse range of mutation types, with the key driver of PC1 being the general difference in the rate of C > G mutations across species, with humans and horses having the highest rate of these changes, and pigs having the lowest. More specific mutational profiles were also observed. For example, pig species appear to present an unusually low rate of C > T mutations in CpG contexts.

Notably, on various principal components, the species do not cluster according to their known evolutionary distance, suggesting that their relationship by mutational profiles is not exclusively a function of their time since divergence. For example, PC5 separates African buffalo from the domesticated bovids, water buffalo and cattle, while PC6 further separates the latter two domesticated species (Supplementary Fig. 2). PC5 is in part driven by African buffalo presenting a comparatively high rate of V[C > A]T changes (Fig. 2b, Supplementary Fig. 3), mutations that are particularly enriched in the COSMIC SBS14 signature associated with defective DNA mismatch repair[25].

To identify the key mutations driving the observed differences between species more clearly, we trained a gradient boosting machine learning model on the SNV mutation-spectra profiles. Consistent with the clear separation in mutation profiles between species, this model was 100% accurate in assigning species labels to individuals (Table 1). From this model, we calculated SHAP (SHapley Additive exPlanations) values[26] that indicated which model features (i.e., mutation type) were most important in differentiating between groups. Figure 2c shows the mutational changes that were most informative in separating species. Notably, only one or two mutation types are sufficient to explain most of the differences between species.

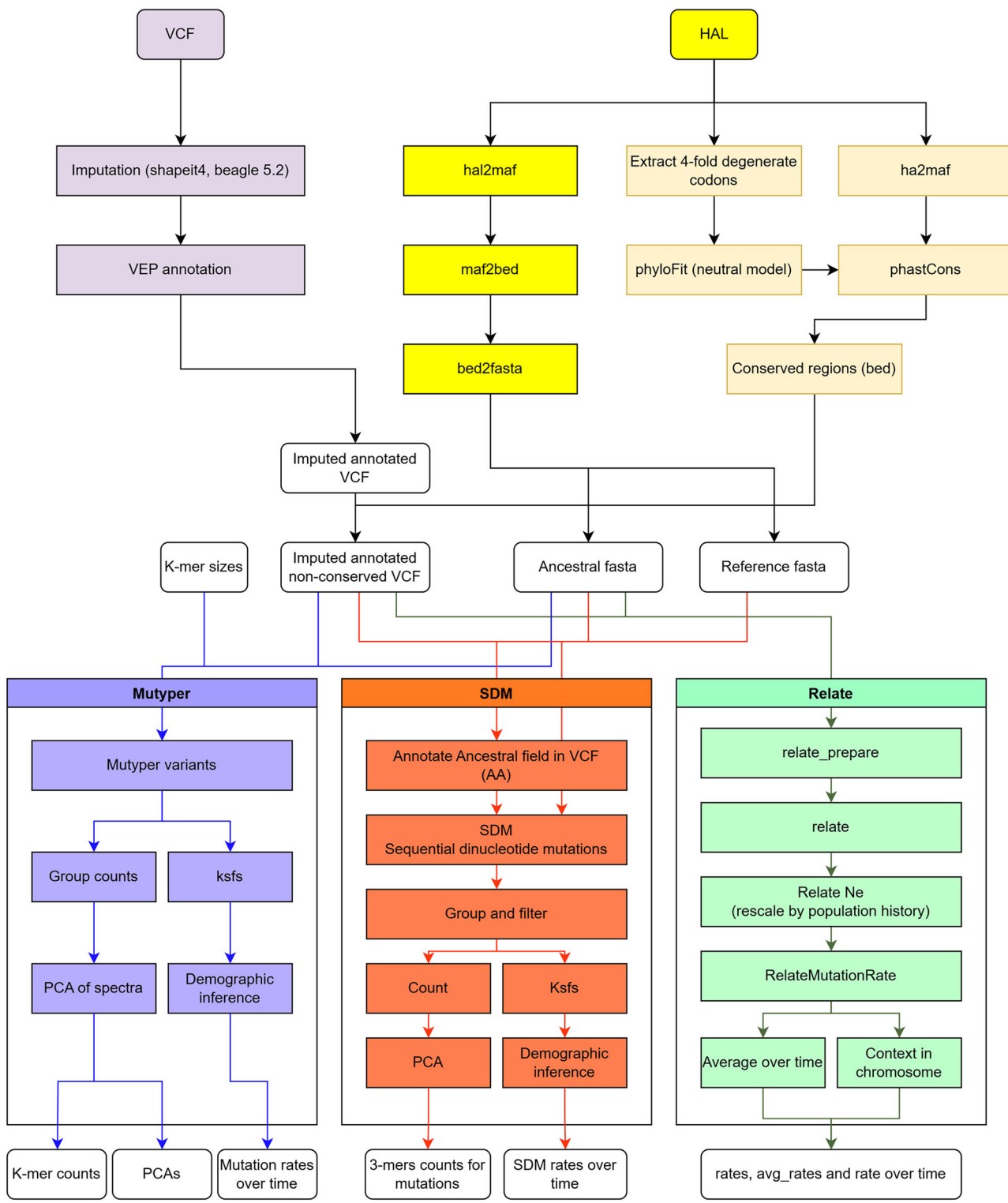

**Fig. 1 | The full nSPECTRa Nextflow workflow.** The workflow is available at www.github.com/evotools/nSPECTRa.

Importantly, these mutational biases are not only observed in intergenic and intronic regions but also among coding changes, meaning that they impact the relative rate of protein evolution between these species. For example, the relative bias for T[A > T]C changes in cattle non-coding regions was also observed among cattle missense changes, skewing the relative rate of amino acid change in this species (Fig. 2d). Additionally, we've found that A[C > T]A changes are significantly different (P-value $< 2.81 \times 10^{-6}$) when comparing bovid with non-bovids species, but not when comparing between bovids (P-value $> 0.24$; Supplementary Fig. 4

and Supplementary Table 1), suggesting this change is specifically enriched and impacting rates of amino acid changes differently among Bovidae. The full list of pairwise comparisons between species with their P-values is shown in Supplementary Data 1.

**SDM mutation profiles identify fine-scale relationships among populations**

Sequential dinucleotide mutations are adjacent single-nucleotide base mutations that are found at different frequencies in the population,

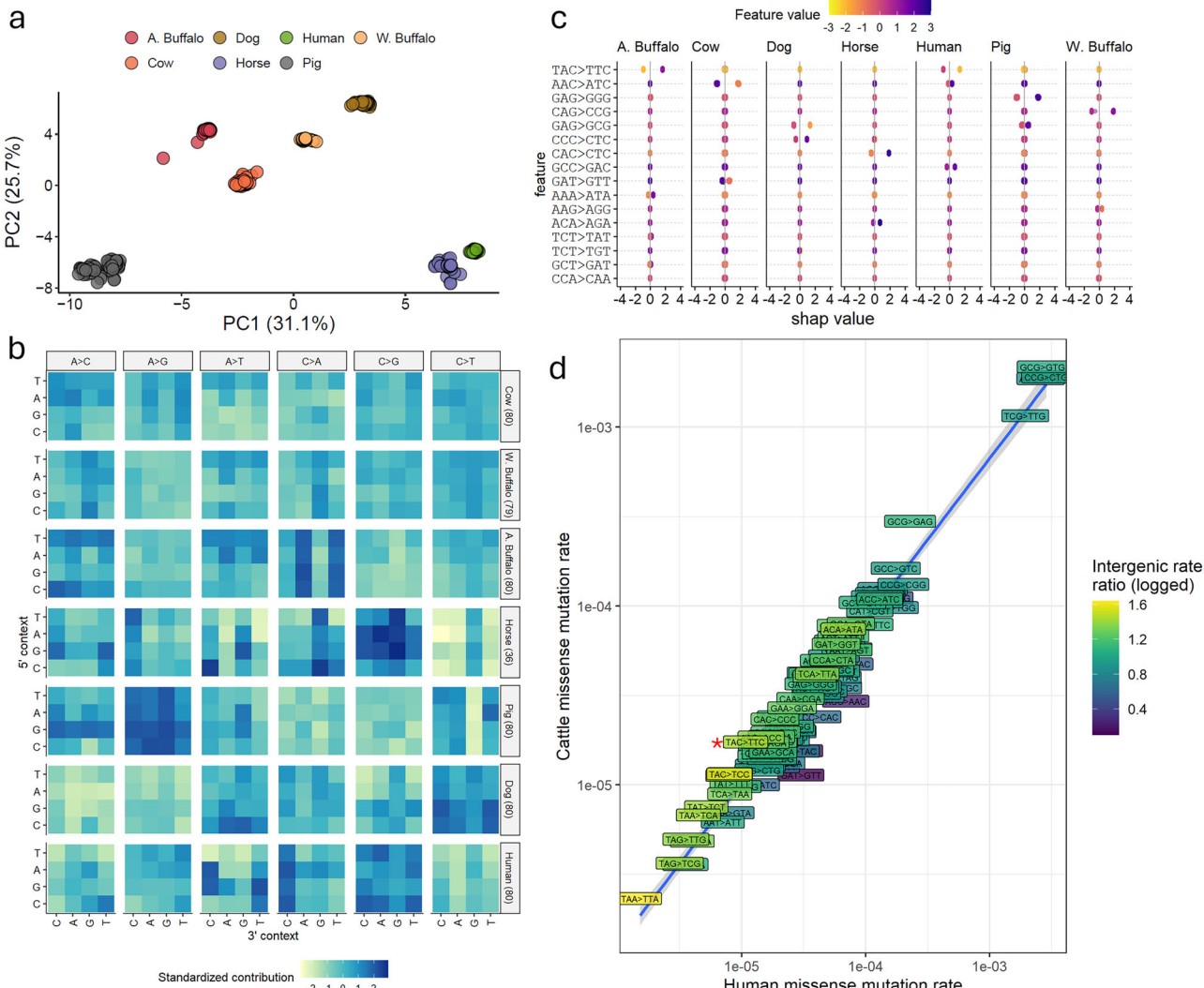

**Fig. 2 | Variations in the SNV rates across species. a** Principal component (PC) analysis of the relationship between different species based on the rate of SNV mutations of different ancestral 3-mers. Each point represents an individual, and the species with larger numbers were randomly downsampled to a maximum of 80 individuals prior to calculating the PCs. **b** Relative contributions of the different mutation types in each species. The 5' and 3' contexts of each change are shown on the left and bottom axes, respectively, with the observed base change at the top. Each row of changes corresponds to a different species, with the downsampled number of individuals shown on the right. Darker blue colours indicate the relative enrichment of the corresponding change. Note that in this plot, the data has been scaled so that each mutation type has a mean of 0 and a standard deviation of 1 across samples. **c** SHAP values associated with changes showing the greatest discrimination among the species. The x-axis indicates the impact of the corresponding feature (y-axis) on the model prediction. A greater spread in SHAP values indicated that the mutation type was more strongly associated with discriminating the corresponding species from others. The colour of the points indicates the feature (mutation) value, and the x-axis indicates whether it is relatively enriched (positive value) or depleted (negative) in the corresponding species. The darker purple positive points indicate that the species is associated with an enrichment of the given change, and the darker purple negative points indicate that the species is depleted with the change. **d** The rate of different changes that lead to amino acid changes in humans versus cattle. Note that the triplet changes correspond to K-mers that may not be in frame with a single codon. The colour of each point corresponds to the ratio of the rate of the same change between the same species, but in intergenic regions. The protein-coding changes enriched in cattle (above the diagonal) are generally also enriched in intergenic regions in cattle (coloured yellow/light green).

suggesting that they are a result of multiple mutational events across different generations. In previous work in humans, we demonstrated SDMs are controlled by distinct mutational processes from SNVs, and that the rate of SDMs in different K-mer contexts can more effectively separate human populations than that of SNVs[6]. This result was observed to carry over across species. Pigs, humans and dogs all separate into sub-groups based on their SDM profiles (Supplementary Fig. 5), with these groups reflecting known evolutionary relationships between sub-populations of these species, namely Asian and European pigs, ancient and modern dog breeds, and African versus non-African human populations. Notably, when analysing mutation profiles within the one species, the Indian water buffalo breeds did not separate with any combination of PCs based on SNV mutational profiles, but were clearly separated on PC2 vs. PC4 calculated from SDM

profiles (Fig. 3). Consistent with this, a gradient boosting model trained on SNV profiles only has an accuracy of 0.13 at assigning correct water buffalo breed labels (Table 1), but in contrast, the accuracy of the model trained on SDM profiles was 0.81. This is despite three of these water buffalo breeds coming from geographically proximal regions of western India and clustering closely on genotype-based PCA (see Dutta et al.)[27]. This illustrates that SDM mutational profiles and machine learning models are potentially effective approaches for assigning breed labels to livestock species.

**Mutation-spectra divergence within breeds and sub-populations**
Cattle are unusual in that they have been domesticated at least twice, leading to two lineages, *Bos taurus taurus* (taurine cattle) and *Bos taurus indicus* (indicine cattle), which last had a common ancestor over 200,000

years ago[28]. These two lineages have historically populated different global areas, with taurine animals migrating to Europe, West Africa, and North Asia, whereas indicine cattle populated South Asia and later East Africa (Fig. 4a). This separation was reflected in their mutation spectra, with the two lineages separating on PC1 for both the SNV and SDM mutation spectra (Fig. 4b, c). However, surprisingly, even animals from the same geographic origin or breed can be further separated by differences in their mutational profiles. West African cattle breeds were separated on PC6 of the cattle SDM mutational spectra, with members of the NDama breed split across the two groups (Fig. 4b), suggesting distinct mutational signatures are segregating within this breed. Intriguingly, Indicine animals from the Indian subcontinent were separated on PC4 of the cattle SNV mutational spectra (Fig. 4c). Notably, African and East Asian indicine animals were similarly separated on this PC. This suggests that two distinct mutational profiles segregate among indicine cattle, with the animals transferred to East Africa carrying a distinct

mutational profile compared to those that migrated to East Asia following the original domestication event of all *Bos indicus* animals on the Indian subcontinent. Examination of the PC4 loadings highlights that the changes linked to the two lineages are specific, with the East Asian indicine lineage comparatively enriched with A[C > T]C and T[C > T]C changes, and the African indicine lineage preferentially associated with N[A > C]G mutations (Supplementary Fig. 7). Sample coverage did not appear to be driving this observation (Supplementary Fig. 8).

### Convergence of mutational profiles across species

Using the SHAP values derived from gradient boosting models applied within species to differentiate population groups, we examined whether certain mutation types are enriched or depleted in populations across species. As shown in Fig. 5a SDM biases are largely all private to subpopulations. This is consistent with SDMs being more specifically associated with particular groups and populations, enabling SDMs to better differentiate between them, and suggests little convergence in SDM profiles across species and populations. However, a range of SNV mutation types were observed to be associated with specific populations across different species (Fig. 5b). In particular, C > T changes in different flanking contexts: for example, the A[C > T]C change is enriched among European humans and depleted from African taurine cattle, and the A[C > T]G change is depleted among European humans, enriched among West African buffalo, and differentiated between ancient and modern dog breeds.

Most notably, one of the main drivers of the separation of East Asian indicine cattle from other indicine cattle is an elevated rate of TCC > TTC mutations. This is the same mutational profile that has been extensively characterised among European humans across a number of studies[5,29], but whose cause is unknown. Closer examination highlights that, as in humans, there has been a pulse of an increase in the rate of these mutations, specifically in this one population of cattle (Fig. 6a). This suggests the same mutational bias has arisen at least twice, once in European humans and once in East Asian cattle. This mutation is a strong contributor to the observed separation of indicine cattle described above on PC4 (Supplementary Fig. 7). Therefore, these results suggest not only are European humans associated with a pulse of TCC > TTC mutations, but this change is specifically linked

**Table 1 | The performance of predicting the population groups within species (or species label across species in the bottom row) using gradient boosting models trained on the spectrum of SNVs or SDMs**

| Populations compared | SNVs | | SDMs | |
|---|---|---|---|---|
| | Accuracy | *P*-Value [Acc > NIR] | Accuracy | *P*-Value [Acc > NIR] |
| Cattle | 0.90 | 2.20E-16 | 0.81 | 2.20E-16 |
| W buffalo | 0.13 | 0.9365 | 0.81 | 5.44E-05 |
| A buffalo | 0.97 | 4.02E-10 | 0.97 | 3.14E-05 |
| Pig | 1 | 2.20E-16 | 0.99 | 2.20E-16 |
| Dog | 0.96 | 6.18E-07 | 1.00 | 8.32E-11 |
| Human | 0.98 | 2.20E-16 | 1.00 | 2.20E-16 |
| Between species | 1 | 2.20E-16 | 1.00 | 2.20E-16 |

One-sided exact test *P*-values of the accuracy above that expected from the no information rate are shown.

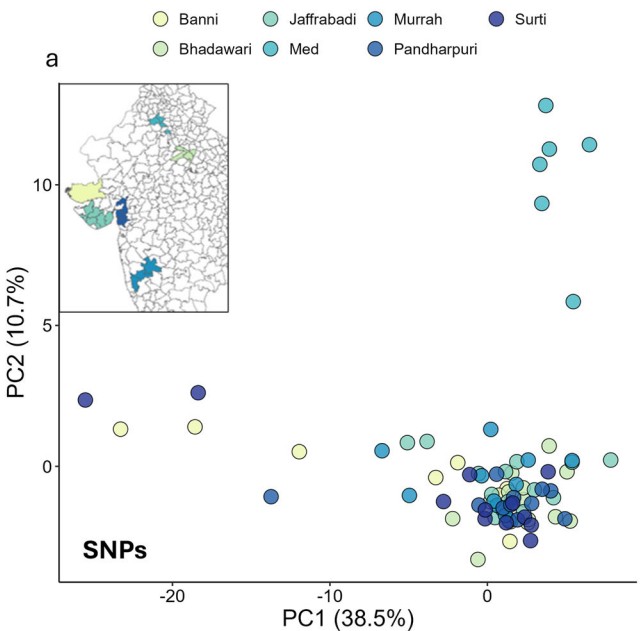

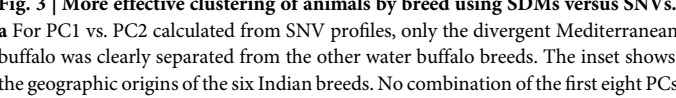

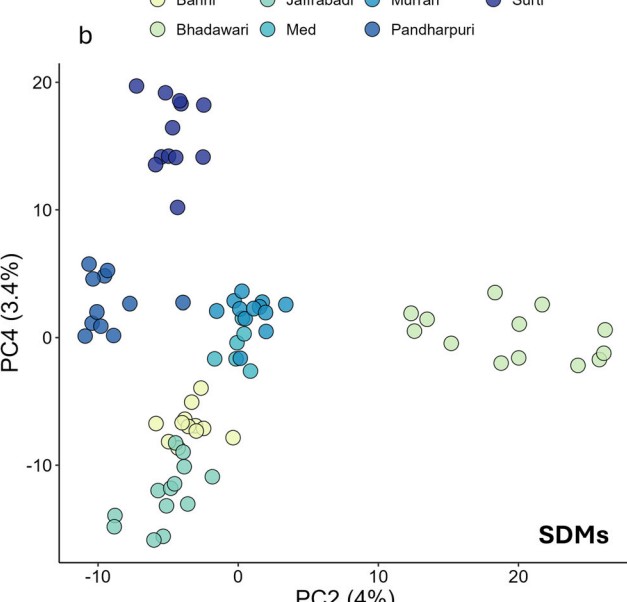

**Fig. 3 | More effective clustering of animals by breed using SDMs versus SNVs.** **a** For PC1 vs. PC2 calculated from SNV profiles, only the divergent Mediterranean buffalo was clearly separated from the other water buffalo breeds. The inset shows the geographic origins of the six Indian breeds. No combination of the first eight PCs calculated from the SNV mutation profiles could be used to separate these breeds. **b** PC2 vs. PC4 based on SDM profiles effectively clusters all seven breeds. PC1 vs. PC2 based on the SDM profiles are shown in Supplementary Fig. 6.

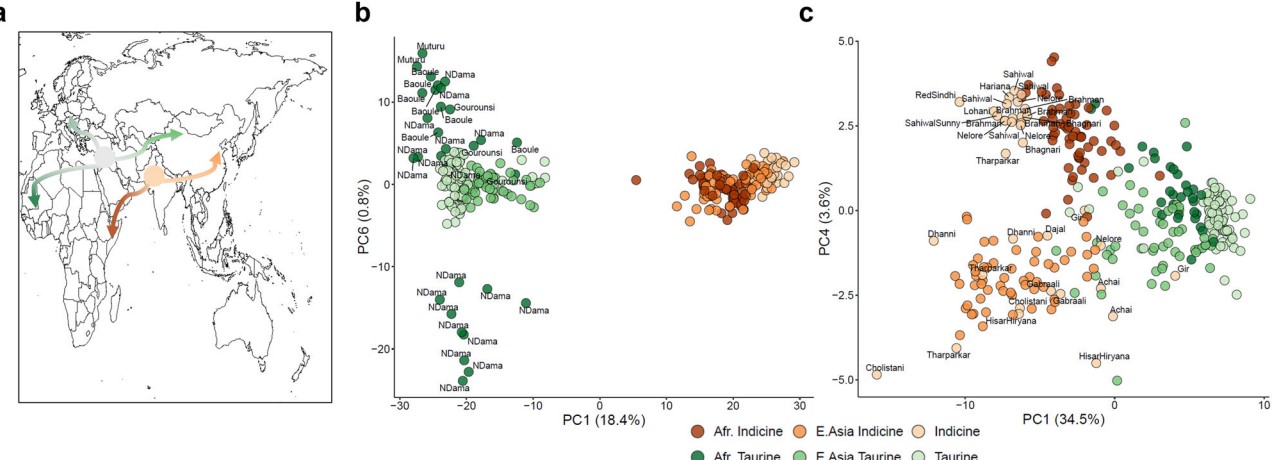

**Fig. 4 | Divergence of mutation spectra within cattle sub-populations. a** The historic migration patterns of the two cattle subspecies, Bos taurus taurus in green and Bos taurus indicus in orange. **b** Separation of West African Bos taurus animals on PC6, calculated from the SDM profiles. Each West African animal was labelled with its breed of origin. The "Taurine" group corresponds to the European taurine animals, and the "Indicine" group corresponds to animals from the Indian sub-continent. **c** The separation of the Indicine cattle on PC4 when calculated from their SNV profiles.

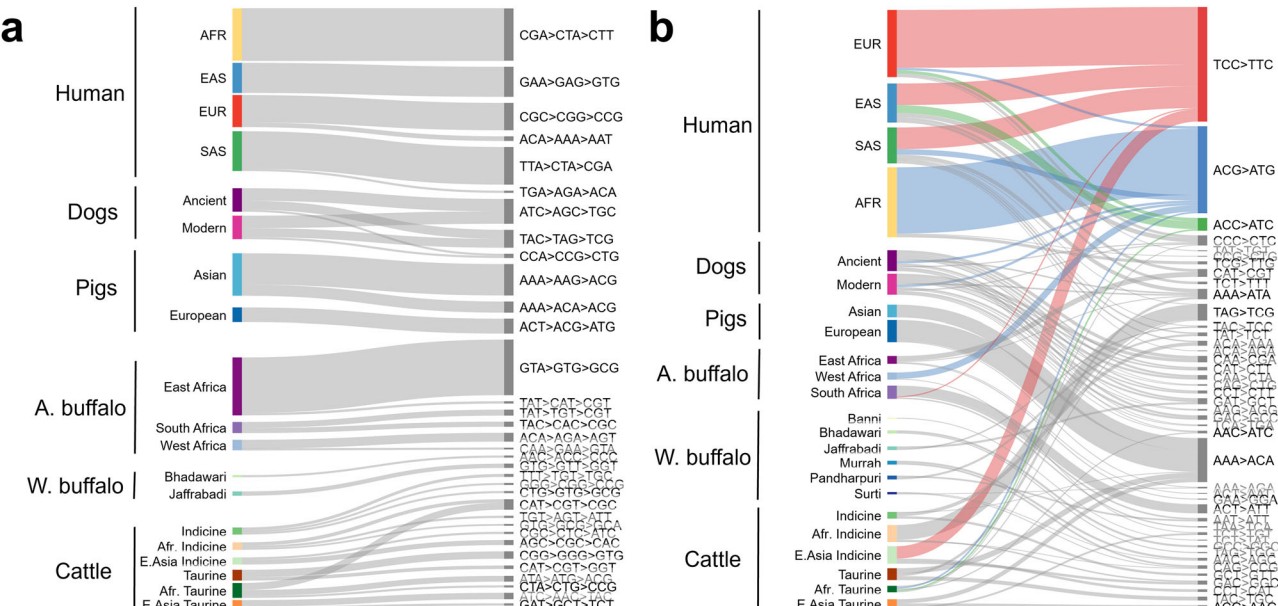

**Fig. 5 | Convergence of SNV mutation profiles across species. a** Discriminating SDM signatures are generally not shared across populations of different species. Lines indicate changes that are preferentially associated with different sub-populations in the SHAP analyses. As the horse had no annotated sub-populations, it is excluded. The width of each bar corresponds to the observed variance in the SHAP values for the corresponding change. The higher the variance, broadly the more strongly the change (shown on the right) is specifically associated with the sub-population (shown on the left). Only changes with a variance greater than 0.05 are shown. **b** In contrast to SDMs, SNV profiles are shared across sub-populations with three selected changes that are linked to sub-populations in different species highlighted.

to a sub-group of indicine cattle that we can map to changes in its frequency between different animals of the same breed.

## Comparison with cancer somatic COSMIC signatures
We finally compared the germline mutational spectra observed in each species to the curated set of human somatic mutational signatures in the COSMIC cancer dataset[30,31], to see if any of the COSMIC signatures may be contributing to the observed mutation profiles. Cosine similarities between the rank 1 nonnegative matrix factorisation (NMF) signatures for each species and the different COSMIC signatures is shown in Fig. 7a. The highest cosine similarities between the signatures for the different species with the COSMIC database signatures range from 0.84 to 0.94 for the horse

and pig species, respectively (Supplementary Table 2), with all of the NMF signatures for the different species most strongly matching the COSMIC SBS5 pattern. This pattern has been observed across several studies to be linked to human germline mutation rates, and this signature appears to make a major contribution to the mutation spectra of all species studied. Across-species links to the different COSMIC signatures were broadly comparable. However, a clear exception is observed for signature SBS1, which is strongly linked to the mutation spectra of all species apart from pigs (Fig. 7a). SBS1 is known to be driven by the deamination of CpG sites to TpG dinucleotides. While this is one of the strongest drivers of mutation rates across all species, it is observed to be comparatively low in pigs (Fig. 7b). To exclude the possibility that a problem with the pig ancestral genome was

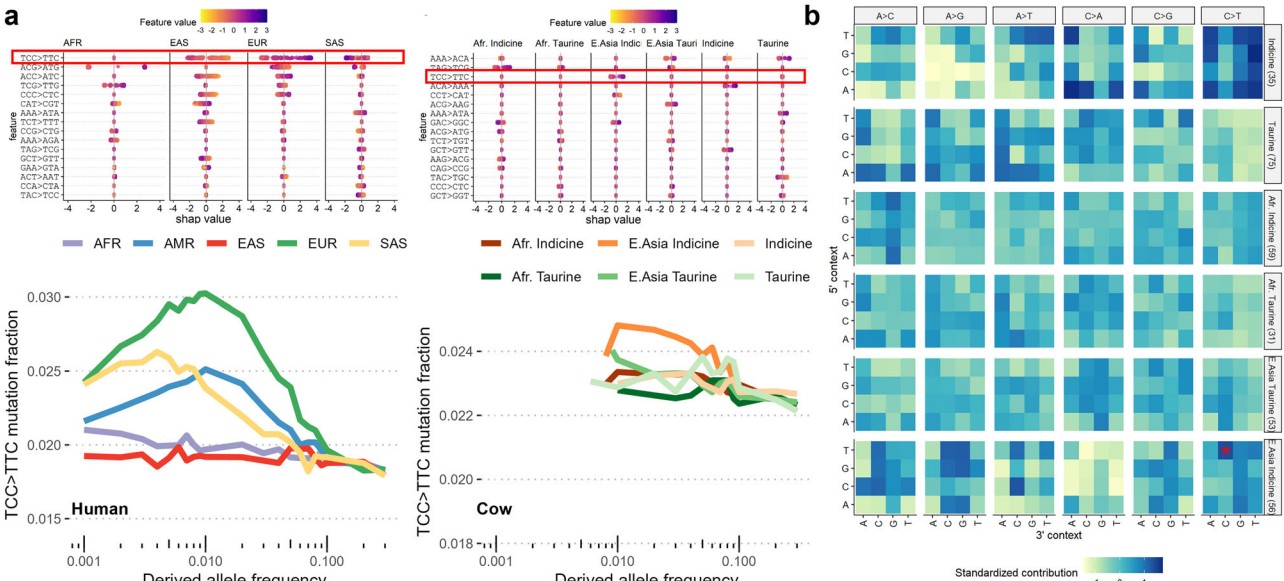

**Fig. 6 | The convergent appearance of TCC > TTC mutation pulses in human and cattle sub-populations. a** The top row shows the SHAP values in human (left) and cattle (right) of the SNV changes most predictive of individual sub-populations. The TCC > TTC change is marked in the red rectangles and associated with separating the sub-populations in both species. The bottom row shows the fraction of TCC > TTC changes found at different frequencies in each sub-population. Consistent with previous observations[5], a pulse is observed in variants with a frequency of around 1% in European humans. A similar enrichment for TCC > TTC changes is observed in East Asian Indicine cattle. **b** The relative enrichment of different changes in the different cattle groups. The TCC > TTC change in East Asian indicine cattle is marked by a red asterisk.

leading us to underestimate the number of changes due to misassigning the direction of change, we collapsed CpG to TpG changes with their reverse change (TpG to CpG). This collapsed frequency was, though, still lower in pigs than the other species examined (Supplementary Fig. 9). Pigs show a generally similar number of methylated CpG sites across their genome as other species[32], meaning a reduced frequency of these sites can also not be invoked to explain this observed difference. Closer examination of the two major pig sub-groups, European and Asian, highlights that Asian pigs are particularly depleted with rare CpG > TpG changes (Fig. 7c), suggesting they are the stronger contributors to this observed depletion.

## Discussion

In this work, we present a workflow designed to streamline the process of characterising different types of mutation spectra in any diploid species. As well as enabling the profiling of the rates of the more commonly studied SNVs, it also enables the characterisation of the rates of SDMs, which we illustrate can provide distinct insights. In particular, SDMs more clearly separate species and are more effective at assigning breed labels to domesticated animals. This is consistent with previous work, specifically in humans, where we illustrated it is the difference in the rate at which the first and second changes occur in SDMs that provides the discriminatory power[6].

The nSPECTRa workflow incorporates all the major steps required to study mutation spectra, from imputation and phasing, through variant annotation and mutation classification using tools like mutyper, to calculating PCs from the resulting mutation profiles. The workflow allows users to run all these steps with little intervention. Importantly, nSPECTRa also incorporates a workflow for reconstructing the ancestral state of a given genome, an important prerequisite for determining the direction of change of mutations. With ancestral genomes currently missing for most species, this can be a major obstacle to performing mutation-spectra analyses. The only prerequisite for determining this ancestral state of a genome is a multiple genome alignment in HAL format containing the reference genome of interest. Some HAL files are publicly available, such as the 241 mammals from the Zoonomia project[33], or can be generated de novo using the progressive cactus alignment tool[34]. Although other studies of mutation spectra have used different approaches to determine ancestral alleles, such as

the *est-sfs* approach[14,35], an important disadvantage of such approaches is that they rely on variant allele frequencies, which are in turn dependent on the availability of good-quality population genomics data for the focal group. In contrast, the approach implemented in nSPECTRa only requires the availability of reference genomes, which are increasingly available for most species, and can characterise ancestral alleles across both polymorphic and fixed sites in the focal species. Consequently, we expect this ancestral genome workflow to be of potential use across studies that require an ancestral sequence and are not restricted to those investigating mutation spectra.

We highlight how the median of ratios method, introduced for normalising RNA-seq data, is also an effective approach for addressing the issue of normalising mutation spectra, so that they can be compared across species that exhibit different mutation counts. An alternate approach adopted in a recent study to correct for this is to downsample the variants to match the species with the lowest diversity[14]. However, as highlighted in this previous study, this method has the potential disadvantage of effectively adding noise to the estimates and can lead to mutation counts of 0 following downsampling, effectively removing signals. Future direct comparison of these two approaches would be of interest to evaluate their relative merits and impact on downstream results.

We also illustrate how the use of SHAP values can effectively identify key mutation types driving differences between species and populations. Downsampling to 30 samples per group results in SHAP values qualitatively similar to the entire dataset, suggesting this approach can be applied even when sample sizes are relatively small. Lowering the sample size further, to 10 individuals per group, renders the SHAP plot uninformative, as shown in Supplementary Fig. 10. Demonstrating the utility of the nSPECTRa pipeline, we characterise the mutation profiles across five domesticated species (pigs, cattle, water buffalo, dogs and horses) and two outgroups (African buffalo and humans) providing insights into the evolution of these species. Cattle were observed to have an unusually complex pattern of mutation-spectra evolution, partly reflecting the fact the species were domesticated at least twice from two independent ancestral populations, leading to distinct mutational profiles in the two lineages. More surprisingly, though, cattle populations that are traditionally treated as single populations separated

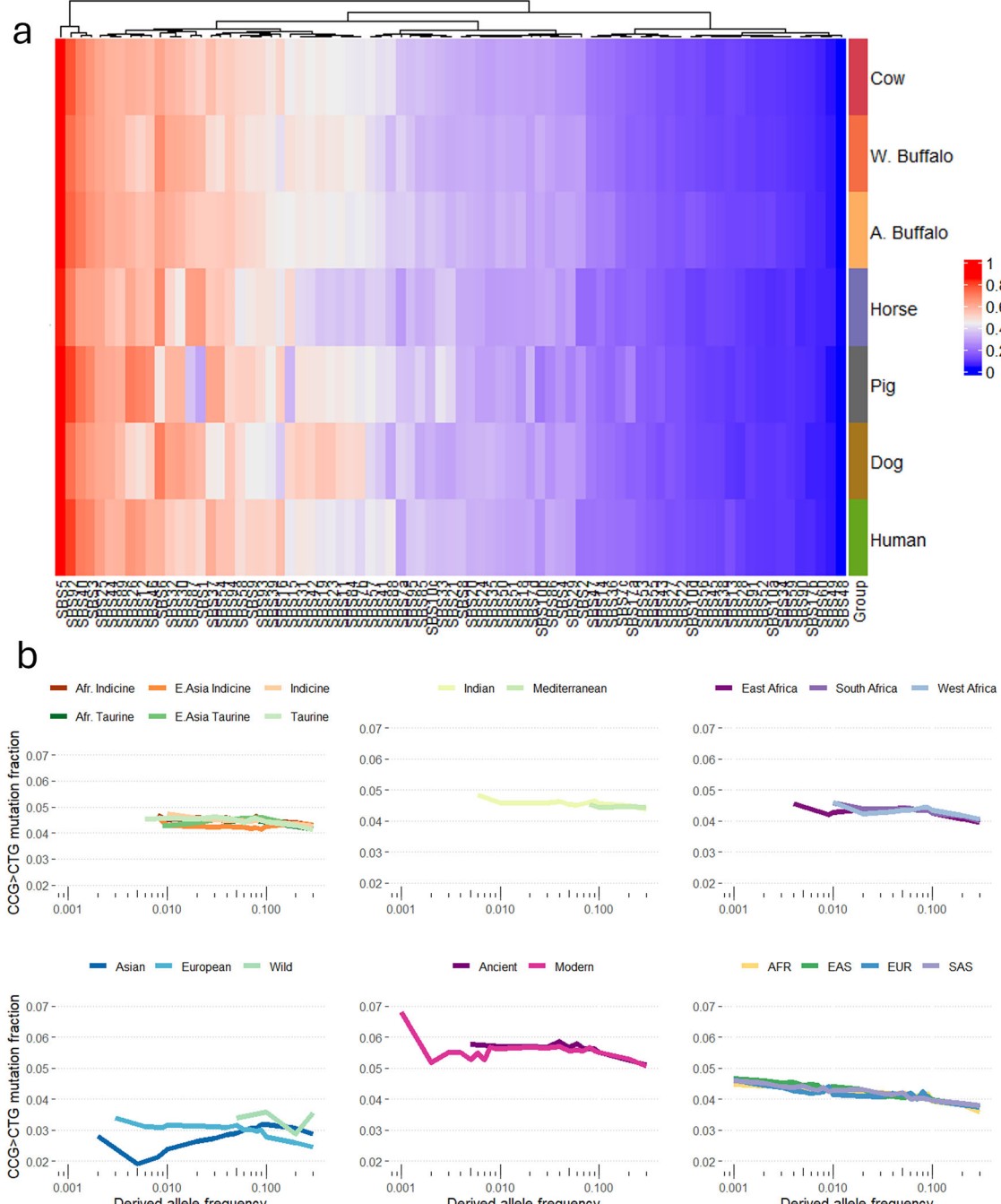

**Fig. 7 | COSMIC signatures in each species. a** Cosine similarities between the NMF signatures on each species and the known COSMIC signatures, with the blue-to-red gradient representative of the 0-to-1 similarity values. **b** The fraction of mutations that are CCG > CTG changes by derived allele frequency across species and populations. The Asian pigs show the lowest rate of these changes, in particular among rare changes. For comparison, data on 15 wild pigs are included in this plot.

based on their mutational profiles. African taurine cattle, including animals from the same N'Dama breed, were separated based on their SDM profiles. The causes of this remain unclear, with no individual SDM change driving this separation, but rather differences across a range of SDM types. It is possible that admixture from other breeds into one or other population could skew their SDM profiles, but arguing against this hypothesis, the samples cluster closely in a traditional genotype-based PCA (Supplementary Fig. 11), which would not be expected if, for example, recent admixture from European cattle had occurred.

Indicine cattle are also separated based on their SNV mutation spectra. In particular, African indicine cattle were associated with N[A > C]G changes and East Asian indicine cattle were linked to A[C > T]C and

T[C > T]C changes, suggesting independent ancestries of these two indicine populations. Most surprisingly, though, indicine animals from the Indian subcontinent could also be broken down into these two lineages based on their profiles, despite their close geographic origin and situation at the original site of domestication. It is possible this is due to a bottleneck event leading to two independent *Bos indicus* lineages, or even potentially two independent Bos indicus cattle domestication events, possibly consistent with the two major mitochondrial haplogroups observed in the subspecies[36]. But arguing against this, the enrichment of TCC > TTC changes observed in East Asian cattle is preferentially found among low-frequency variants, suggesting the divergence in the rate of these changes happened more recently. Perhaps more plausibly, introgression from other Bos species

may have driven these differences, with up to 1.57% of Chinese indicine genomes proposed to have arisen through introgression from banteng, *B. javanicus*[37]. Future studies intersecting the location of such regions with known DNA repair genes, or the study of the mutational spectra in *B. javanicus*, may provide insights into the potential causes of these distinct mutational spectra.

A further potential avenue of future research is using nSPECTRa to investigate the sex chromosomes, which were excluded in the current analysis. For example, investigating whether male and female germ cells are associated with different mutation spectra.

Another interesting addition to the analyses would be to expand the investigation to closely related wild species. Species such as the wild Yak (*Bos mutus*) and the Chimp (*Pan troglodytes*), two species with high-quality reference genomes and large sequencing projects, would potentially offer further insights into the evolution of mutational profiles across multiple mammalian species and following domestication.

A particularly notable finding is the observation of the pulse in TCC > TTC changes, previously observed in European humans, in East Asian cattle. Although strongest for this change, this convergence in mutational profiles across populations from different species is seen for other mutation types and suggests expanding these analyses to further species and populations would identify more examples of such convergence in mutation spectra. This could potentially be used to refine the potential mechanisms underlying shared biases in mutation spectra.

These differences in mutation spectra are expected to have downstream consequences. For example, we demonstrate that the biases in mutation rates in non-coding areas of the genome are also observed in coding regions, consequently impacting the rate of change of different amino acids. This ultimately means that not all amino acid changes are equally likely across species, and certain species may be prone to accumulate certain deleterious mutations at different rates over extended timescales, potentially influencing factors such as codon usage bias[38].

Some limitations of the current study should be highlighted. Notably, when comparing mutation spectra across species, different genomic regions are invariably studied. Although the results are normalised by sequence content, and the majority of the genome was studied in each species, this could lead to mismatches in the types of sequences examined. For example, certain repeat classes are not found in all species[39], which may have distinct mutational profiles. However, other approaches to try and mitigate this issue, for example, restricting the analyses to orthologous regions found across every species, would also likely lead to biases such as towards ultra-conserved regions. For this reason, we have primarily focused on the within-species analyses. A further aspect to note is the fact that the population levels differ between species: for instance, cattle and water buffalo are broken down by breed, while other populations are by geographical area. This is partly a result of available annotations for the different datasets and partly because the concept of breed does not apply to species such as the African buffalo. But this means the ability to separate populations should not be directly compared across species. Likewise, changes in mutation spectra can have multiple causes, such as changes in demography. This is potentially the driver of the observed difference in CpG deamination rates in pigs. A previous study in human populations highlighted that differences in the rate at which CpG sites mutate is a result of the fact that these sites are more likely to mutate more than once. This impacts their relative frequency, and the occurrence of this is dependent on demography[29]. Furthermore, while early and foundational mutation-spectra studies generally counted variants in each carrier individually, one proposed approach when undertaking principal component analysis of mutation spectra is to randomly allocate each variant to a single carrier[14,19]. This method can potentially help reduce inflated similarities due to shared ancestry or close relatedness. However, it also discards potentially important information concerning the overall mutational load of individuals, and the composition of the dataset (e.g., which samples are included) impacts the variants assigned to each sample and consequently their assayed mutation spectra. Hence, while random

allocation may reduce over-clustering in certain scenarios, it also has the potential to introduce dataset-dependent biases.

In conclusion, we introduce the nSPECTRa workflow for determining mutation spectra in any species, and we illustrate the utility of approaches such as the median of ratios method to normalise the output of this workflow. We also demonstrate their utility by applying them across seven different mammalian species to study the evolution and convergence of mutation spectra. With the increasing availability of population genetic data for an ever-expanding set of species, we expect these resources and approaches to be of widespread use to provide further important insights into the evolution of mutation spectra and their relevance to shaping species and populations.

## Methods
### Defining the ancestral sequence
A reliable definition of mutation changes relies on the accurate definition of the ancestral state of each base in the genome of interest. To achieve this, we used CACTUS[34] which can generate ancestral genomes from multiple genome alignments. For the dog, human, horse and pig species, we provided the 241-way cactus alignment generated by Armstrong et al.[34] and accessible at https://cglgenomics.ucsc.edu/data/cactus/. For each species, we specified a different ancestral branch, representative of the different clades: fullTreeAnc209 for the dog, fullTreeAnc105 for the human, fullTreeAnc92 for the macaque, fullTreeAnc226 for the horse and fullTreeAnc192 for the pig.

As this existing alignment did not contain suitable genomes for the bovids, for these we generated a cactus alignment containing 17 publicly available genomes (Supplementary Table 3). The supporting phylogenetic tree was defined using the MASH (v2.2) software[40] with default parameters to calculate the pairwise genome distances. These were then used as input for Phylip's neighbour algorithm, using the UPGMA approach, to define the guide tree for the CACTUS run. This phylogenetic tree was represented using FigTree[41] and shown in Supplementary Fig. 12.

Prior to the generation of the multiple genome alignments for the 17 assemblies, these were masked using a combination of DustMasker[42], WindowsMasker[43] and RepeatMasker[44] with trf[45], to provide homogenous and comparable levels of soft-masking of the different genomes.

Cactus (v2019.03) was then used to generate the multiple genome alignments and the ancestral state at each split of the phylogenetic tree. The ancestor Inner 13 has been selected as the ancestral state for all the *Bovinae* genomes, since common among all of them. This Inner 13 ancestral genome presents a good coverage of all the assemblies. And for the cattle, water buffalo and African buffalo genomes, we had coverages of 90%, 94% and 93.5%, respectively.

### Datasets and variant pre-processing
We considered variant datasets for 7 different species in this study: human, cattle, water buffalo, African buffalo, horse, pig and dog.

The human dataset was generated as part of the 1000 Genomes Project, and consists of 3202 samples from five super-populations[46]. The cattle (*Bos taurus*) dataset consists of 477 animals from 71 worldwide populations, representative of both subspecies (*B. t. taurus* and *B. t. indicus*) and from four continents (Europe, Asia, Africa and South America) described in Zhao et al.[47]. The water buffalo (*Bubalus bubali*) dataset consists of 79 water buffalo genomes of 7 breeds including the Mediterranean buffalo from Italy and 6 others from India[27]. The African Buffalo (*Syncerus caffer*) dataset consists of 196 buffaloes from 4 different subspecies (*S. c. caffer*, *S. c. nanus*, *S. c. aequinoctialis* and *S. c. brachyceros*) from conservation areas across the range of this species in Africa[48]. The pig (*S. scrofa*) and horse (*E. caballus*) datasets were retrieved from the Genome Variation Map database[49].

The human genomes had already been filtered and phased. For the other species, we performed the same pre-processing of the datasets, reducing as much as possible factors such as batch effect and depth of sequencing effects. Any cattle, African buffalo, macaque, horse and dog samples with very high missingness (>75%) and low average depth of

sequencing (<8×) were discarded. For pigs, only the high missingness filter (>75%) could be applied due to the absence of the DP flag in the VCF annotation. For each dataset we only retained biallelic SNPs with a call rate > 90% and minor allele count (MAC) > 2. Following variant filtering, we also retained only 2nd-degree-unrelated individuals identified by converting the biallelic SNPs to plink binary format using plink v1.90b4[50] and then running the KING software[51] (v2.2.4) with options --unrelated --degree 2. After excluding the related individuals, we performed a second round of variant filtering to only retain variants with a minor allele count >2 and a missing frequency <10%. After all the filtering the number of samples for each dataset was as follows: 309 cattle and 64,377,549 variants; 175 African buffalo and 41,226,919 variants; 79 water buffalo and 24,057,736 variants; 36 horses and 15,956,650 variants; 350 pigs and 27,402,978 variants; 606 dogs and 15,750,570 variants; and 2561 humans and 49,890,585 variants.

Following the definition of the ancestral state, the workflow extracts the sites for which we could successfully define the ancestral allele (i.e., either the reference or the alternate allele matches the ancestral allele) and with derived allele frequency (DAF) < 0.98. This retained 26,908,970 variants for the cattle; 18,342,327 variants for the African buffalo; 10,371,926 variants for the water buffalo; 7,427,557 variants for the horses; 11,986,618 variants for the pigs; 7,319,094 variants for the dogs; and 18,383,907 variants for the humans.

### Determining mutation counts

The inference of the mutation spectra was performed using Mutyper (v0.6.1) and the sequential dinucleotide mutation (SDM) approach from Prendergast et al.[6]. The two algorithms were implemented into nSPECTRa (nextflow SPECTRum analysis), a Nextflow mutation spectrum analysis workflow that performs the analyses in a species-agnostic, low-interaction and highly parallelised fashion. The Relate software, that can calculate mutation rates and their change over time[22], was also included as an option in this workflow, but was not used in the current study. The scheme of the workflow can be seen in Fig. 1, and can be summarised as follows: (1) pre-processing of the VCF file; (2) preparation of the ancestral reference (3) computation of the mutation spectra.

The nSPECTRa VCF pre-processing consists of first imputing the sparse missing genotypes, followed by phasing. The workflow supports either Beagle v5 or greater[52] or shapeit4 v4.2.0[53]. In this study, we applied Beagle (v5.2) with the following effective population sizes (Ne): 1000 for the cattle[27], 358 for water buffalo[27], 5000 for the African Buffalo[54], 5000 for the pig[55], 100 for horse[56] and 230 for dog[57].

Following the imputation and phasing, the workflow uses the variant effect predictor (VEP) software to define the effect of each variant considered in the analysis. This step can be performed by either passing a cache compliant with the installed version of VEP[58], or by providing a custom GFF, suitable for recently annotated genomes or genomes for which a cache is not available. More specifically, we used the VEP cache v104 to annotate the cattle and the dog datasets (references ARS_UCD1.2 and CanFam3.1, respectively), VEP cache version 97 for the human (reference GRCh38.p12), and 94 for the domestic Horse (reference EquCab2.0), the NCBI GFF annotation for the water buffalo and the pig (reference Sscrofa 10.2) and the Ensembl rapid release annotation for the African buffalo (reference GCA_902825105.1).

The ancestral reference state consists of detecting the most likely ancestral base in the reference genome used to call the variants. To do so, nSPECTRa takes the HAL generated by CACTUS as input, and extracts the alignments from the reference and the Inner 13 genome in multiple alignment format (MAF) using the hal2maf function from HAL[59]. The alignments are then processed using a series of custom scripts that identify the most likely base in the Inner 13 genome covering the reference genome, and uses this information to generate an ancestral reference genome with the ancestral base instead of the reference base, where defined. For non-Bovidae species, the principle is the same, but uses a different set of alignments and different ancestral genomes for each species: fullTreeAnc209 for the dog,

fullTreeAnc105 for the human, fullTreeAnc92 for the macaque, fullTreeAnc226 for the horse and fullTreeAnc192 for the pig.

Once the inputs are prepared, the third stage can be run using one of the three software previously described. For Relate, the workflow runs all the steps required on the guide, preparing the inputs, estimating the effective population size and, finally, calculating the mutation spectra and mutation rate.

For mutyper, the software can calculate the mutation spectra for all samples, chromosome by chromosome to increase the throughput, for a set of K-mer sizes specified by the user, allowing them to account for multiple context sizes (e.g., $K = 3$ will consider 1 basepair flanking sequences such as A[T > C]A, whereas $K = 5$ will consider 2 basepair flanking sequences such as TA[T > C]AG). The raw mutation counts are then normalised using the initial state frequency in the ancestral sequence (e.g., the number of A[T > C]A changes is divided by the number of ATA K-mers in the ancestral genome). This allows the user to factor in the increased or decreased probability of a mutation occurring because of the intrinsic K-mer content of the ancestral genome. If mutyper and relate are run together, the Ne will be used in combination with the Ksfs vector generated by mutyper to perform demographic analysis and inference on the different populations considered.

Finally, SDM will estimate the rate of sequential dinucleotide mutations for each individual in each population, considering not only the context but also the effect of the variants and their position in the codon. The raw counts are normalised for the initial K-mer content in the ancestral sequence, analogously to what is done for the Mutyper counts.

The minimum requirements to run nSPECTRa are a VCF file containing genotypes for one or more samples and either an existing ancestral genome or a suitable HAL file so that one can be calculated de novo. Note that the calculation of SDMs requires population-level data. Running the whole nSPECTRa workflow on the largest (human) dataset of 3202 individuals, including generation of a new ancestral genome, characterisation of the individual changes and SDMs, required 1358 CPU hours.

### Grouping of the populations

The grouping of the different individuals has been done according to the individual characteristics of each dataset considered.

For the cattle, we classified the individuals as belonging to 6 major groups: African indicine ($n = 59$), E.Asian indicine ($n = 56$), Indicine ($n = 35$) East Asian Taurine ($n = 53$), African Taurine ($n = 31$) and taurine ($n = 75$). Indicine refers to indicine animals from the Indian subcontinent, the site of original indicine domestication, and taurine refers to taurine breeds of European origin.

Water buffaloes were classified into the seven breeds described in Dutta et al.[27]: Banni ($n = 12$), Bhadawari ($n = 13$), Jaffrabadi ($n = 13$), Murrah ($n = 12$), Pandharpuri ($n = 11$), Surti ($n = 12$) and Mediterranean ($n = 6$).

African buffaloes have been classified depending on their macro-geographical origin: Eastern African ($n = 101$), Western African ($n = 36$), Southern African ($n = 38$).

The pig dataset has been categorised depending on the individual geographical ancestry: European ($n = 161$), Asian ($n = 180$) and wild pig species ($n = 15$). This separation was validated by a PCA of the samples. Seven samples that were extreme outliers on this PCA were excluded from downstream analyses.

The horse dataset presents the lowest number among the species considered ($n = 36$) and lacks information on any sub-groupings; therefore, these samples were considered as a single pan-population.

Due to their high degree of diversity, dogs have been classified by applying the hierarchical clustering on principal component (HCPC) method from the FactoMineR R package[60]. This identified two groups: modern breeds ($n = 507$), and ancestral breeds and village dogs ($n = 117$).

Finally, humans are classified into the same 5 super-populations, as defined by the 1000 genome project: East Asian ($n = 506$), South-East Asian ($n = 504$), European ($n = 524$), African ($n = 678$) and American ($n = 349$)

**Article**

but due to their high levels of admixture the American population was dropped from analyses.

Following normalising SNV and SDM mutation counts by the occurrences of their ancestral K-mer in the ancestral genome, we then applied the median of ratios method[24] to enable the comparison of normalised counts across individuals and species. We implemented the median of ratios as an R script function that applies the formula described by Anders S. and Hubers W. (2010):

$$\hat{s}_j = median_i \frac{k_{ij}}{(\prod_{v=1}^m k_{iv})^{1/m}}$$

First, we compute the geometrical mean for each change across all samples. We then divide the value at each count for each sample by the geometrical mean of each count type. Then, we compute the median ratio for each sample, producing a normalisation factor. Finally, each original count is divided by the normalisation factor, obtaining the final value to be used for the analyses. Due to the larger number of zeros in the SDM matrix for the human species, the median of the ratio can return infinite normalised values for samples with more than half the values equalling 0. To avoid this problem, we added a small value to the K-mer-normalised counts. This small number is equal to 1 multiplied by $10^N$, where N is the decimal notation of the smallest value in the matrix (e.g. if the smallest value is $3.5 \times 10^{-9}$, we add $1 \times 10^{-9}$ to each K-mer-normalised value).

Dividing the values by the K-mer counts in the ancestral genome allows us to make the datasets comparable across species by accounting for the probability of a change occurring in a species due to an increased number of sites. The second normalisation allows us to make samples comparable when accounting for different numbers of mutations across species.

### Cross-species spectra analysis

We investigated the relationships between the populations for both SNVs and SDMs using a PCA on the normalised values. We represented the heatmap of the single-change mutation matrix pooled for each population using a modified version of the *plot_standardized_profile_heatmap* function from the MutationalPatterns R package[61], and represented the single-change mutation profile using a modified version of the *plot_96_profile* function from the same package.

For the multispecies comparisons, we first combined all the K-mer normalised datasets, and then normalised them through the same median of ratios approach. We represented the relationship among the mutation spectra in the different species using the same methods previously described.

Derived allele frequencies in the different populations have been computed using the bcftools +fill-tag command, and values were then extracted using bcftools query. These have then been processed by a custom script that counts how many rare changes occur in each group.

Changes showing significant differences in their frequencies (Supplementary Table 1 and Supplementary Data 1) were identified using the method described by Harris and Pritchard[5].

### Modelling and SHAP values

To identify which mutation types best differentiate populations and species, we fitted the normalised mutation spectra as features in multi-class extreme gradient boosting models using the xgbtree method in the caret R package[62] with five-fold cross-validation. To assess model performance, 20% of the individuals were left out of the data prior to model training to act as an independent test set from which model metrics such as accuracies could be derived.

To identify those genetic changes that best differentiate populations, we calculated SHAP[26] values from these models using the xgboost R package[63].

### Comparison with COSMIC profiles

Finally, we compared the spectrum for each species with the COSMIC signatures (v3.3)[31]. The signatures for the GRCh38 human genome have been downloaded directly from the COSMIC website (https://cancer.sanger. ac.uk/signatures/documents/2047/COSMIC_v3.3.1_SBS_GRCh38.txt) and processed locally.

We identified the mutation signatures by applying the NMF analysis implemented in the R NMF package[64] (rank = 1 and performing 10 iterations). After extracting the signatures, we identified the most similar SBS COSMIC profile by using a customised version of the rename_nmf_signatures function from the MutationalPatterns R package, which computes the cosine similarity between the NMF and the COSMIC profiles. We used a cosine cut-off of 0.8 to identify the most similar signature in the COSMIC database.

After identifying the most similar COSMIC signatures, we extracted the contribution to the signatures for each species.

### Reporting summary

Further information on research design is available in the Nature Portfolio Reporting Summary linked to this article.

## Data availability

The human 1000 genomes cohort genetic variants were obtained from https://www.internationalgenome.org/data-portal/data-collection/30x-grch38, The pig (S. scrofa) and horse (E. caballus) datasets were retrieved from the Genome Variation Map database (https://ngdc.cncb.ac.cn/gvm/), the dog genotypes from https://sra-pub-src-1.s3.amazonaws.com/SRZ189891/722g.990.SNP.INDEL.chrAll.vcf.1 and the cattle and water buffalo genotypes were those published in Dutta et al.[27]. The African buffalo data from Talenti et al. 2023 is available at ENA under accession PRJEB59220. The mutational profiles for the different species have been deposited in Zenodo with https://doi.org/10.5281/zenodo.15276107[65]. The data underlying Figures is provided as Supplementary Data 2.

## Code availability

The Nextflow workflow (nSPECTRa) is available at https://github.com/evotools/nSPECTRa and in Zenodo, with https://doi.org/10.5281/zenodo.10784677.

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

## Acknowledgements
The authors would like to thank Professor Albert Tenesa and Dr Ismail Ozkaraca for helpful discussions regarding the manuscript. This work was supported by BBSRC (Grant Nos. BB/T019468/1 and BBS/E/RL/230001A).

## Author contributions
J.G.D.P. conceived the study, A.T. and J.G.D.P. designed the analyses, and A.T. and J.G.D.P. performed them. L.J.M. contributed to the conceptualisation of the study. T.W. contributed to the preparation of the datasets.

## Competing interests
The authors declare no competing interests.
