## [Transparent Peer Review file · Communications Biology]

The evolution and convergence of mutation spectra across mammals

Corresponding Author: Professor James Prendergast

Version 0:

Reviewer comments:

Reviewer #1

(Remarks to the Author)

Summary:

The study by Talenti et al. provides a comprehensive analysis of germline mutation spectra across various mammalian species, focused on domesticates. The authors developed a novel workflow, nSPECTRa, to characterise mutational spectra and applied it to seven mammalian species. The study reveals significant divergence in single nucleotide and dinucleotide polymorphism mutation profiles across species and identifies patterns of convergent evolution in mutation rates. The research highlights the complexity of mutation spectra within breeds and sub-populations, particularly in cattle. The paper does a good job of highlighting some of the challenges inherent in interpreting these types of data, particularly how demographic processes, such as population structure, can impact the data. The authors have made their analysis pipeline available in nextflow and I anticipate that this may be of great use to those working on comparative genomics and mutational processes.

I think the paper makes a valuable contribution to the field. In particular it uses an innovative methodology for comparing mutational spectra across species (the use of SHAP values). I congratulate the authors on their study which is innovative in its methodology and provides novel biological insights. I have some minor comments and requests for clarification from the authors.

The use of SHAP values by the authors to compare spectra is innovative and could potentially be a valuable contribution to the field. To demonstrate the power of using SHAP values and encourage their wider adoption it might be helpful if the authors provided a supplementary analysis comparing the results using SHAP values in comparison to using other approaches (e.g. downsampling) to compare mutational profiles on their dataset. Can they demonstrate that the SHAP values capture information that other approaches fail to detect? This is not essential, just a suggestion to consider.

Relatedly, to encourage others to adopt the nSPECTRa workflow it could be helpful if the authors provide some benchmarking information on how computationally intensive/ time consuming the different stages are to run. This would help other groups assess if they have the computational infrastructure to run this workflow with their datasets.

Given that the complex recent demographic histories of domesticates could be impacting their mutational spectra and the title of the paper is more broadly about mammals I found it surprising there were not more non-domesticated species included in the analysis. I expect this may be due to the wider availability of population wide data from domestic species. This is reasonable but it does limit the inferences that can be drawn about the evolution of mutation spectra more broadly across mammals.

The paper utilises population datasets to study the evolution of germline mutational spectra between species. This population data enables the authors to study the allele frequency spectrum. However for species where population data is not available but a single genome is, would their pipeline still be of utility to compare germline mutational spectra between species? If so this could be valuable for scientists working on species where population data is not yet available. Essentially how different are the inferences that can be drawn from comparing fixed differences between species from those using population data? Emphasising the value of population data over single genomes could strengthen the manuscript.

Related to this it would be interesting to see a comparison of how the fixed derived germline spectra in a species (obtained using ancestral allele inference to the species reference genome) compares to the population level data the authors have

generated. This could address if the mutational spectra of more recent germline mutations (those that have not yet reached fixation) is similar to older germline mutations (those that are fixed) in the species. This could perhaps give an indication of how conserved/mutable the germline mutational spectra is in a species. Though I am sure the authors will have thought about this more deeply and there may be reasons such an analysis would not be appropriate.

Similarly it might be of interest for the authors to provide some indication of how many individuals are necessary to yield insights into differences in germline mutational spectra by doing a subsampling analysis and seeing if the same core findings could have been achieved using a smaller number of samples. This may give some indication of how sample size influences the findings which could guide future studies. This is not essential.

It is an interesting observation that SNV patterns are shared between populations but SDMs are far more discriminatory (Relating to Fig 5). I would be interested to hear why the authors think this is the case. Is it because SDMs likely represent two independent processes that have sequentially occurred causing SNV changes and the chance of both processes occurring in two populations is low? Or do the authors imagine other explanations? Further discussion of the potential biological basis of these observations could be interesting.

Germline mutations predominantly arise in the male or female germ cells and there is evidence that the mutational spectra varies between sexes. Did the authors consider trying to tease apart these spectra, for example by comparing mutational spectra between the X and the autosomes, as the X spends only $\frac{1}{3}$ of its time in males versus half for the autosomes? This may be beyond the scope of this paper but would be an interesting analysis if there was sufficient data (the number of mutations on the X chromosome may still be insufficient for a meaningful comparison).

Did the authors consider doing a de-novo signature extraction on their datasets, as is commonly done for cancer datasets using methods based on non-negative matrix factorisation? This may help to identify the relative activity of the mutational processes that are active within species. Or perhaps the authors explored this and found it did not add anything beyond their PCA based approach?

The SDM results are very interesting. Are the authors confident that these dinucleotide mutations are not the result of sequencing artefacts? I wonder if indels that have fixed between breeds that are not present in the reference genome could perhaps be leading to mismapping errors. Are SDMs enriched in repetitive sequences? If this is the case the SDMs could still be real but it would be good to hear that the authors thoroughly investigated these mutations and are satisfied that they are not likely to be artefactual.

The PARP11 GWAS result is intriguing. Did the authors check if there are sequence data available from cells treated with PARP inhibitors that show similar spectra? For example the Signal dataset has some data from cells treated with Olaparib (PARP inhibitors) <https://signal.mutationalsignatures.com/explore/experimentalSample/143>. It doesn't look to me that the detected mutations match PC4 but this is an avenue the authors might consider exploring.

The authors mention that the depletion of CpG mutations in the pig may be due to recent demographic changes. Is there evidence that the pigs sampled or domestic pigs in general have gone through more extreme recent demographic changes than bovinds and other domesticates that would be consistent with these results? The potential impact of demography on these results seems to be one of the main limiting factors for clear interpretation of the data (a common and unavoidable issue in population genomics).

Reviewer #2

(Remarks to the Author)

This paper presents an interesting analysis of mutation spectrum variation in a collection of mostly-domesticated animals, as well as a release of the pipeline they used to streamline these analyses. The mutation spectrum comparisons certainly contain interesting signals to follow up on, and the pipeline could be useful for democratizing this style of analysis.

My major critique of the paper is that it does not do much to tire-kick its methods and results for potential bioinformatic confounders and statistical problems. In some cases, I have critiques of specific methodological choices and will explain how these could have led to misleading conclusions. In other cases, my concern is that there were no routine checks for potential problems, and I will describe why there is good reason to do these checks.

Major comments:

When performing principal component analysis on mutation spectrum data, it is best practice to randomly allocate each SNP to just one of the individuals who carries the derived allele. From my read of the methods, I wasn't able to confirm that this was done (apologies if this was done and I missed it). For example, if a particular derived allele has allele count 10, it should be counted toward the mutation spectrum of just one of those 10 haplotypes, chosen at random. If instead the allele is counted toward the mutation spectra of all 10 carrier haplotypes, this will inflate the appearance of mutation spectrum similarity among individuals who are more genetically similar. I don't expect this random subsampling to affect the extent of mutation spectrum divergence between species as far apart as pig and human, since pig and human share very few derived alleles, but I do expect it to increase the variance within pig and within human so that the extent of clustering on the PCA plots will only reflect mutation spectrum differentiation and will not reflect the fact that certain pairs of animals simply share more derived alleles than others. If this random subsampling was not done, this might make the SDM spectrum PCAs

especially misleading, since the state space of SDMs is larger than the state space of single trinucleotide mutations and each SDM category likely has fewer mutational observations in it. If a large proportion of the SDMs in a given indicine cow are shared with other indicine cows and a smaller proportion are shared with taurine cows, this could substantially inflate the appearance of SDM spectrum differentiation between these populations. As a thought experiment, you can imagine simulating some data where the mutation spectrum is the same across all subpopulations but you only sample a fairly small number of mutations compared to the total number of mutational categories. If most of your SNPs are shared between two or more individuals from the same population, it will look like your different populations have different mutation spectra even when you know that this is not true.

I'm concerned about the validity of the association study in Figure 4 because its setup and assumptions are very different from those of a conventional GWAS, and it seems really susceptible to confounding by population structure. The reason it's so different from a conventional GWAS is that the mutation spectrum of each cow in the study is really a weighted average of the mutation spectra of a bunch of the cow's ancestors (since the data are polymorphisms rather than de novo mutations). The reason that mutation spectrum PCAs show signal is because of population structure: each cow's ancestors tend to be more similar to one another than to some other cow's ancestors, at least when that other cow is from a different population. So the presence of population structure is essential to the premise of being able to identify associations with the PC4 mutational spectrum. In a normal GWAS, however, your analysis is hinging upon an assumption of the absence of population structure given that population structure could create a bunch of spurious associations with your trait, which is the reason for controlling for the first 20 genotype PCs. Spurious associations caused by population structure are already a serious problem for normal traits like height when they are strongly differentiated between subpopulations, even when investigators control for lots of genotype PCs. I unfortunately can't suggest a quick fix that would be able to test whether this analysis is trustworthy, but my sense is that proving this kind of GWAS is statistically valid would be enough work that it would need its own dedicated methods paper where you rigorously simulate an association between a mutation spectrum gradient and a causal allele, ascertain mutation spectra the way you are doing here, and test whether you are able to identify the correct locus using this pipeline. The discussion gives the caveat that the PARP11 association still needs to be replicated in additional data, but this seems like way too weak a caveat—it would be appropriate if this were a regular GWAS where you were directly measuring a regular phenotype in each individual. There is enough in the paper that it could still merit publication without this association test, so removing it and going on to write a methods paper proving its validity could be the best way to go.

The quality filters that are imposed on the SNPs in this paper are quite lax: SNPs are only required to have 8x sequencing depth and less than 75% missing data. How were these quality thresholds chosen, and what is the evidence that they are good enough? I recognize that it isn't realistic to hold non-model-organism data to the same standards that would be realistic for human data, but since there hasn't been any investigation of the possible effects of low data quality, I'm concerned that the paper's results might be substantially affected by bioinformatic error. Some of the early reports of mutation spectrum differences between human populations were later shown to be artifactual (Anderson-Trocme, et al. *Mol Biol Evol* 2020), so it is a real possibility that more serious artifacts could create some of the patterns in the current paper, especially when a lot is being made of the distribution of higher order PCs. One realistic quality check to implement would be to focus on the paper's main results (e.g. the TCC signature and PC4 gradient in cattle, the CpG transition deficit in pigs) and check whether these patterns still hold if the QC thresholds are strengthened to require say 20x depth and less than 20% missing data. The reproducible workflow that the authors have made would benefit a lot from the addition of some automated quality checks, and one could be a quantification of how much the k-SFS changes when the quality thresholds are varied. In particular, I noticed that there aren't any checks of the quality of the ancestral state identification being performed, which will depend a lot on how well the people running the pipeline have chosen the outgroup genome. Even in a best case scenario, the site frequency spectrum usually has a bit of a "smile" shape that indicates there are too many SNPs with frequencies close to 1—can the authors add a check of this 'smile' to the pipeline so that researchers will be aware if the ancestral state identification is not working very well? It is also best practice to discard SNPs above say a frequency of 90% since most of these SNPs will have misidentified ancestral alleles and will have the potential to confound the mutation spectrum.

The presence of excess TCC>TTC mutations in the indicine cattle is a cool result, but I think the paper overstates the strength of the evidence that this is the same as the mutational signature found in European and South Asian humans. The reason I say this is that the SHAP value plots don't seem to indicate an excess of the minor components of the European TCC pulse, including CCC>CTC, ACC>ATC, and TCT>TTT. There are at least two other examples of distinct mutational signatures that are also dominated by TCC>TTC but which have different minor components: the COSMIC UV light signature SBS18 and a signature found in the canine transmissible venereal tumor (Baez-Ortega, et al. *Science* 2019) where the TCC>TTC mutations have a biased pentanucleotide context distribution that is not seen in humans.

Additional minor comments:

In line 59, it doesn't make sense to refer to SDMs as being caused by two 'independent' mutation events—if the first mutation increases the probability that the second mutation will occur, these are two statistically dependent events. It would be more accurate to say that an MNP is caused by a single concerted mutation while an SDM is caused by two mutations occurring at different times.

Lines 71-72 state that CpG>TpG mutations make up over a third of all observed mutations, and a third seems too high. I couldn't find where this figure came from when I skimmed the cited 2005 paper. If you look at Figure 2a of a more recent paper, Johnson, et al. *Nature* 2017, you see that CpG>TpG mutations are less than 20% of human germline de novo mutations. It would be better to specify what species you're talking about if you want to stand by the 1/3 figure, and whether you are talking about de novo mutations or polymorphisms.

I don't think lines 91-93 accurately represent Bergeron, et al.'s interpretation of their finding of elevated mutation rates in domesticated animals. As I understand it, Bergeron, et al. argued that population bottlenecks in domesticated animals may have relaxed selection on mutator alleles, leading to increases in these species' mutation rates. This still counts as a change in these species' underlying mutational processes—without such a change, there's no reason that a historical

population bottleneck could impact the rate or spectrum of de novo mutations.

Some of the references are cited as preprints when published versions are available—the authors should update these as needed.

Lines 100-102 say that “little work has been done to study mutation spectra differences outside of the major model organisms due to a lack of suitable large whole-genome sequenced cohorts and easy-to-use software tools.” Arguably the two most relevant papers to cite here would be Beichman, et al. 2023 and Bergeron, et al. 2023, which are not the two papers actually cited. (Bergeron, et al. were not able to study context-dependent mutation spectra but still looked at 1-mer mutation spectrum differences between vertebrate classes). I also think it comes across a bit strangely to say that “little work” has been done when in this case nonzero work has been done and it’s a bit subjective to say whether the amount is little or big. As an alternative, the authors could either briefly describe what has been done or not done or else make some more precise claim about what they do that previous work has not done. In terms of why there has not been more work on mutation spectra in non-model organisms, I think the authors are missing what is perhaps the most important reason: non-model organism sequences tend to be of more variable bioinformatic quality than human and mouse sequences, so we have to worry more about reporting mutation spectrum differences that are really bioinformatic artifacts.

Can the authors verify whether different filters based on SNP density might have been applied to the datasets they are using? Some bioinformatic pipelines discard clustered SNPs that might support MNPs or SDMs, so it is important to verify that this is not creating any artifactual SDM patterns.

The intro points out that Bergeron et al. did not test whether domesticated species have a different mutation spectrum from non-domesticated species, effectively implying that this current paper will fill that gap. However, this paper doesn’t really address that question either, which would require pairing each domesticated species with a closely related non-domesticated species and testing whether there is some systematic difference between the domesticates and non-domesticates. It’s fine that this paper doesn’t tackle this question, but it seems misleading to imply that they are doing this by pointing out that Bergeron et al. did not do it, rather than just saying that the Bergeron results imply that domesticated species’ mutation spectra might be interesting and you wanted to learn more about them.

Reviewer #3

(Remarks to the Author)

The manuscript entitled “The evolution and convergence of mutation spectra across mammals” authored by Talenti et al., describes a new workflow, nSPECTRa, to characterize mutation spectra. They demonstrate the power of the approach investigating the evolution of mutations in trinucleotides in domestic animal species and human. Their results are quite interesting, as they demonstrate differences between species, between and within populations, but also cases of convergent evolution between species.

I found the manuscript clear and very well written. It was a pleasure reading it. Here are my observations and suggestions:

1) The differences observed result from the different efficiency with which DNA repair mechanisms correct specific mutations. I wonder if some variation in these genes can be identified by association analyses. This should be possible particularly in the case of differences in mutation rates observed within a single population (e.g. N'Dama cattle). When happening in a same genetic background the effect is likely due to rather simple genetic variation that should be simpler to identify.

2) The divergence in mutation spectra within population is intriguing. My first interpretation was in favour of a trait not so important to estimate diversity (e.g. red and black Holstein differ in MC1R gene and coat colour but they are and will remain a single population), however there is a more profound effect, as with time DNA repair genes variation leads to profound genetic divergence. What appears to be a single event (SNV), starts a domino effect and with time induces MNPs and divergence. Can this be claimed as contributing to speciation in the long term?

3) Line 69. Selection, as well as drift, influences the distribution of allele frequencies in populations, but it’s not clear how it influences the rate of mutation of some sequence. Can the authors clarify? I imagine they refer to the fact selection favors beneficial and purges deleterious mutation but this is not affecting “the rate by which certain sequences are prone to mutate”. The mutational profile changes, but the rate of mutation should not.

4) line 120. The ability to infer ancestral allele on nSPECTRa would benefit from validation, e.g. using information on wild relatives of domestic species or ancient DNA data publicly available.

5) Figure 2b. In this figure I can’t identify the high frequency of C>T transversions in CGC trinucleotides (CGC TGC or CGT), as the deamination of methylated cytosines in CpG island is very frequent. Has this kind of mutation been left out of the analyses?

6) lines 416-419. Have authors considered ancient contribution of wild relative species?

7) lines 426-428. The hypothesis of two indicine domestications or at least two subgroups also results from mtDNA variation.

8) lines 459-460. Does this also justify the species-specific codon usage?

Version 1:

Reviewer comments:

Reviewer #2

(Remarks to the Author)

I appreciate the thorough revisions that the authors have undertaken—these have substantially improved the manuscript and I only have a few minor remaining comments.

On the subject of randomly allocating each variant to one individual before performing a PCA, it's true that this might not have been done in the earliest mutation spectrum papers, but it's described in the methods of some more recent papers that rely heavily on mutation spectrum PCA (e.g. Goldberg and Harris 2022 <https://doi.org/10.1093/gbe/evab104> as well as Beichman, et al. 2023 <https://doi.org/10.1093/molbev/msad213>). In their response, the authors point out some legitimate disadvantages of this approach, including that changes the balance of rare and common alleles and it means that the mutation spectrum assigned to an individual depends on which other individuals are being jointly analyzed. In my view, these disadvantages are outweighed in the context of PCA analyses where the point is not to ascertain each population's mutation spectrum, but to test whether patterns of mutation spectrum similarity and divergence are sufficient to assign individuals to populations. The authors say that it isn't really a confounder if two individuals have similar mutational loads due to shared ancestry, but would they see it this way if two individuals in the dataset were siblings, in which case more than half of the mutations making up each sibling's mutation spectrum would be shared with the other sibling's mutation spectrum? Based on such a PCA, you would likely conclude that these two siblings had more similar mutation spectra than two unrelated individuals from the same population, even if the spectra of the siblings' non-shared mutations were as divergent as the spectra of two unrelated individuals. This sibling example is intentionally extreme, but each pair of unrelated individuals from the same species do often share many common genetic variants. For this reason, I'd suggest giving the approach a try or at least including a discussion section caveat saying that this random variant allocation wasn't done, meaning that shared variants might inflate some of the observed clustering patterns.

The authors also point out some complexities that are important to consider when formulating allele frequency cutoffs. I think the option they implement for discarding high frequency alleles assuages my concerns well. The only point in the response I disagree with is that smile plots are less useful when dealing with domesticated or otherwise bottlenecked populations. Even though bottlenecks will increase the rate of fixation of derived alleles, they will also increase the rate of loss of derived alleles, so a smile pattern should not be present unless caused by bioinformatic errors or widespread positive selection.

I appreciate the clarification regarding the Bergeron high mutation rates issue, and I think specifying that the text is referring to per-year mutation rates would be helpful (I was confused because I assumed the statement was referring to per-generation mutation rates).

Reviewer #3

(Remarks to the Author)

Authors have answered by questions and accepted some of my suggestions. I am happy with this version of the manuscript.

RESPONSE TO REVIEWERS

We would like to express our genuine gratitude to the reviewers for their extremely helpful and constructive comments on our manuscript which we sincerely believe have substantially strengthened the analyses. We outline below how we have addressed each of the reviewers' comments. We believe we have managed to successfully address each point but please let us know if you believe there is anything we have missed.

Reviewer #1 (Remarks to the Author):

Summary:

The study by Talenti et al. provides a comprehensive analysis of germline mutation spectra across various mammalian species, focused on domesticates. The authors developed a novel workflow, nSPECTRa, to characterise mutational spectra and applied it to seven mammalian species. The study reveals significant divergence in single nucleotide and dinucleotide polymorphism mutation profiles across species and identifies patterns of convergent evolution in mutation rates. The research highlights the complexity of mutation spectra within breeds and sub-populations, particularly in cattle. The paper does a good job of highlighting some of the challenges inherent in interpreting these types of data, particularly how demographic processes, such as population structure, can impact the data. The authors have made their analysis pipeline available in nextflow and I anticipate that this may be of great use to those working on comparative genomics and mutational processes.

I think the paper makes a valuable contribution to the field. In particular it uses an innovative methodology for comparing mutational spectra across species (the use of SHAP values). I congratulate the authors on their study which is innovative in its methodology and provides novel biological insights. I have some minor comments and requests for clarification from the authors.

The use of SHAP values by the authors to compare spectra is innovative and could potentially be a valuable contribution to the field. To demonstrate the power of using SHAP values and encourage their wider adoption it might be helpful if the authors provided a supplementary analysis comparing the results using SHAP values in comparison to using other approaches (e.g. downsampling) to compare mutational profiles on their dataset. Can they demonstrate that the SHAP values capture information that other approaches fail to detect? This is not essential, just a suggestion to consider.

Apologies, we may be misunderstanding this comment, and please let us know if we are, but we don't think SHAP values and downsampling are alternative approaches. We primarily used SHAP values to disentangle the mutation changes that best differentiate between populations. The majority of changes show some degree of differentiation between groups but by using SHAP values we can identify those changes that best distinguish them. Importantly these changes are largely non-redundant, so it reduces the large number of changes to a small number, ranked by importance. In contrast, if for example you look at PCA loadings (for example as in Supplementary Figure 3), there can be a large number of changes with high/low loadings and these strictly are associated with PCs not populations.

The downsampling approach we mention in the paper has been used in previous studies to put mutation spectra on the same scale, whereas we proposed the use of the median of ratios approach to do this (SHAP values being calculated from these normalised data either way). We did this as downsampling has the issue that you are effectively throwing away data, increasing noise in the estimates and potentially reducing a range of counts all to 0 which before downsampling had different levels.

Relatedly, to encourage others to adopt the nSPECTRa workflow it could be helpful if the authors provide some benchmarking information on how computationally intensive/ time consuming the different stages are to run. This would help other groups assess if they have the computational infrastructure to run this workflow with their datasets.

Thank you for the suggestion. We have now included timings for running the whole workflow on the largest (human) dataset of 3202 individuals, including generation of a new ancestral genome and running the mutyper and SDM workflows, at line 627-630. The workflow required approximately 1,400 CPU hours for the largest dataset without imputation, or around 150 CPU hours for the smaller African buffalo dataset with imputation.

Given that the complex recent demographic histories of domesticates could be impacting their mutational spectra and the title of the paper is more broadly about mammals I found it surprising there were not more non-domesticated species included in the analysis. I expect this may be due to the wider availability of population wide data from domestic species. This is reasonable but it does limit the inferences that can be drawn about the evolution of mutation spectra more broadly across mammals.

As the reviewer suggests we had focused predominantly on domesticated (mammalian) species due to the availability of population data. These seven species do span a substantial range across eutherian mammals and as two of the seven species studied are not domesticated (humans and African buffalo) we felt we couldn't really use the term "domesticated mammals" in the title. However, we do try and highlight early in the abstract the particular focus on domesticated species: "*We apply nSPECTRa to seven species, including several that have undergone domestication*" (lines 18-19) and why domesticated species may be unusually interesting mammals to study (lines 14-16).

The paper utilises population datasets to study the evolution of germline mutational spectra between species. This population data enables the authors to study the allele frequency spectrum. However for species where population data is not available but a single genome is, would their pipeline still be of utility to compare germline mutational spectra between species? If so this could be valuable for scientists working on species where population data is not yet available. Essentially how different are the inferences that can be drawn from comparing fixed differences between species from those using population data? Emphasising the value of population data over single genomes could strengthen the manuscript.

Yes this pipeline can in theory be run across one individual to get its single nucleotide mutation spectra. However, interpretation of the results may be challenging in that the extent to which results represent an individual, population or species would be unclear. As the definition of SDMs requires population frequency data SDMs results wouldn't though be usable. We have now highlighted this at lines 625-627.

Related to this it would be interesting to see a comparison of how the fixed derived germline spectra in a species (obtained using ancestral allele inference to the species reference genome) compares to the population level data the authors have generated. This could address if the mutational spectra of more recent germline mutations (those that have not yet reached fixation) is similar to older germline mutations (those that are fixed) in the species. This could perhaps give an indication of how conserved/mutable the germline mutational spectra is in a species. Though I am sure the authors will have thought about this more deeply and there may be reasons such an analysis would not be appropriate.

We agree with the reviewer that understanding how mutation rate in a species has changed over time is important and have tried to address this point by comparing the mutation profile across different allele frequencies, with variants with a high derived frequency being generally older and about to reach fixation. For example, see bottom panels in Figure 6A. Importantly, as the reviewer suggests, the mutation spectra does appear to change over time.

Similarly it might be of interest for the authors to provide some indication of how many individuals are necessary to yield insights into differences in germline mutational spectra by doing a subsampling analysis and seeing if the same core findings could have been achieved using a smaller number of samples. This may give some indication of how sample size influences the findings which could guide future studies. This is not essential.

Thanks for the comment. This is a slightly difficult question to answer as the impact depends on the samples being studied and the particular analysis being undertaken. For example, when comparing mutation spectra between species, the main differences in mutation spectra are observed even when restricting to ten samples per species. For example the samples still separate on a PCA and the mutation spectra heatmap is highly similar to the original shown in Figure 2B:

However, the SHAP plots are uninformative with this dataset being too small for effective modelling:

When one reaches a sample size of 30 the SHAP results broadly reflect the original results shown in Figure 2C in terms of the informative features:

We have now added a section to the discussion describing this (lines 416-420).

It is an interesting observation that SNV patterns are shared between populations but SDMs are far more discriminatory (Relating to Fig 5). I would be interested to hear why the authors think this is the case. Is it because SDMs likely represent two independent processes that have sequentially occurred causing SNV changes and the chance of both processes occurring in two populations is low? Or do the authors imagine other explanations? Further discussion of the potential biological basis of these observations could be interesting.

We showed in previous work that the discriminatory power of each of the two individual changes that make up an SDM is actually largely the same as for SNVs (*Prendergast JGD et al. Genome Biol Evol. 2019 Mar 1;11(3):759-775*). It is only when you start looking at the frequency at which neighbouring changes co-occur that the increased separation is observed. We showed if you calculate the difference in frequencies between first and second changes in SDMs this is also effective at separating populations. So it is this difference in the rate at which first and second changes occur in different kmer contexts between populations that effectively provides the discriminatory power. We have now added some text to the discussion referring to this (line 383-385).

Germline mutations predominantly arise in the male or female germ cells and there is evidence that the mutational spectra varies between sexes. Did the authors consider trying to tease apart these spectra, for example by comparing mutational spectra between the X and the autosomes, as the X spends only $\frac{1}{3}$ of its time in males versus half for the autosomes? This may be beyond the scope of this paper but would be an interesting analysis if there was sufficient data (the number of mutations on the X chromosome may still be insufficient for a meaningful comparison).

In our current analyses, we had excluded the sex chromosomes, in part because of the issues the reviewer highlights. However, this is an interesting idea, and we have highlighted how this would be interesting to investigate in future analyses at line 452-454.

Did the authors consider doing a de-novo signature extraction on their datasets, as is commonly done for cancer datasets using methods based on non-negative matrix factorisation? This may help to identify the relative activity of the mutational processes that

are active within species. Or perhaps the authors explored this and found it did not add anything beyond their PCA based approach?

We believe the type of analysis the reviewer is referring to is shown in Figure 7A? This did help identify specific cosmic signatures enriched among the mutation profiles.

The SDM results are very interesting. Are the authors confident that these dinucleotide mutations are not the result of sequencing artefacts? I wonder if indels that have fixed between breeds that are not present in the reference genome could perhaps be leading to mismapping errors. Are SDMs enriched in repetitive sequences? If this is the case the SDMs could still be real but it would be good to hear that the authors thoroughly investigated these mutations and are satisfied that they are not likely to be artefactual.

We thank the reviewer for the suggestion. We downloaded the same version of the reference genome used for the VEP annotation, matching release for each dataset (with the exception of the African buffalo, for which annotation information are only partial). We then extracted the masked portions of the genomes using twoBitInfo -maskBed, and intersected these with the SDM sites identified for each species. The maximum overlap of SDMs with repetitive regions was in cow at 11.6%. So in all species the overwhelming majority of SDMs are not found in repetitive regions. Furthermore, the workflow currently extracts the masked regions from the genomes included in the graph automatically, and automatically splits the SDM file into within and outside the softmasked regions found in the genome so users can compare the results.

The PARP11 GWAS result is intriguing. Did the authors check if there are sequence data available from cells treated with PARP inhibitors that show similar spectra? For example the Signal dataset has some data from cells treated with Olaparib (PARP inhibitors) <https://signal.mutationalsignatures.com/explore/experimentalSample/143>. It doesn't look to me that the detected mutations match PC4 but this is an avenue the authors might consider exploring.

This is an interesting idea but unfortunately we couldn't find much data on the mutation spectra observed following the use of PARP inhibitors. The paper to which the reviewer refers actually did not find a mutational signature linked to Olaparib above background (Table S2 of Kucab et al.), suggesting many of the mutations in the shared link are actually likely due to culture conditions. In accordance with reviewer 2's suggestion we have now removed this analysis from the manuscript.

The authors mention that the depletion of CpG mutations in the pig may be due to recent demographic changes. Is there evidence that the pigs sampled or domestic pigs in general have gone through more extreme recent demographic changes than bovinds and other domesticates that would be consistent with these results? The potential impact of demography on these results seems to be one of the main limiting factors for clear interpretation of the data (a common and unavoidable issue in population genomics).

All domesticated species have had complex evolutionary histories, often involving multiple bottleneck or founder events. So teasing out aspects specific to pigs that may be driving this would likely be tricky. However, although not presented in this paper, we have subsequently been involved in looking at mutation spectra in goat populations,

where a similar depletion was observed, suggesting this phenomena may not actually be restricted to pigs.

Reviewer #2 (Remarks to the Author):

This paper presents an interesting analysis of mutation spectrum variation in a collection of mostly-domesticated animals, as well as a release of the pipeline they used to streamline these analyses. The mutation spectrum comparisons certainly contain interesting signals to follow up on, and the pipeline could be useful for democratizing this style of analysis.

My major critique of the paper is that it does not do much to tire-kick its methods and results for potential bioinformatic confounders and statistical problems. In some cases, I have critiques of specific methodological choices and will explain how these could have led to misleading conclusions. In other cases, my concern is that there were no routine checks for potential problems, and I will describe why there is good reason to do these checks.

Major comments:

When performing principal component analysis on mutation spectrum data, it is best practice to randomly allocate each SNP to just one of the individuals who carries the derived allele. From my read of the methods, I wasn't able to confirm that this was done (apologies if this was done and I missed it). For example, if a particular derived allele has allele count 10, it should be counted toward the mutation spectrum of just one of those 10 haplotypes, chosen at random. If instead the allele is counted toward the mutation spectra of all 10 carrier haplotypes, this will inflate the appearance of mutation spectrum similarity among individuals who are more genetically similar. I don't expect this random subsampling to affect the extent of mutation spectrum divergence between species as far apart as pig and human, since pig and human share very few derived alleles, but I do expect it to increase the variance within pig and within human so that the extent of clustering on the PCA plots will only reflect mutation spectrum differentiation and will not reflect the fact that certain pairs of animals simply share more derived alleles than others. If this random subsampling was not done, this might make the SDM spectrum PCAs especially misleading, since the state space of SDMs is larger than the state space of single trinucleotide mutations and each SDM category likely has fewer mutational observations in it. If a large proportion of the SDMs in a given indicine cow are shared with other indicine cows and a smaller proportion are shared with taurine cows, this could substantially inflate the appearance of SDM spectrum differentiation between these populations. As a thought experiment, you can imagine simulating some data where the mutation spectrum is the same across all subpopulations but you only sample a fairly small number of mutations compared to the total number of mutational categories. If most of your SNPs are shared between two or more individuals from the same population, it will look like your different populations have different mutation spectra even when you know that this is not true.

We are not aware of the suggested practice of just assigning a mutation randomly to a single individual for this type of analysis. For determining the SNV mutation spectra we used the same approach and tool released by the authors of one of the foundational papers looking at mutation spectra in this way (*Harris K, Pritchard JK. Rapid evolution of the human mutation spectrum. Elife. 2017 Apr 25;6:e24284*). The SDM analysis was also carried out as previously described (*Prendergast JGD et al. Genome Biol Evol. 2019 Mar 1;11(3):759-775*). In these analyses we are characterising the mutational load of individuals, and if two individuals have similar mutational loads because of shared

ancestry, we don't see this as being a confounder. One issue of randomly assigning mutations to just one individual is that this analysis will be confounded by factors such as the frequency of the allele or the size and makeup of the analysed population. For example, the number of mutations assigned to an individual would depend on how many other individuals it is processed alongside, introducing a novel confounder.

I'm concerned about the validity of the association study in Figure 4 because its setup and assumptions are very different from those of a conventional GWAS, and it seems really susceptible to confounding by population structure. The reason it's so different from a conventional GWAS is that the mutation spectrum of each cow in the study is really a weighted average of the mutation spectra of a bunch of the cow's ancestors (since the data are polymorphisms rather than de novo mutations). The reason that mutation spectrum PCAs show signal is because of population structure: each cow's ancestors tend to be more similar to one another than to some other cow's ancestors, at least when that other cow is from a different population. So the presence of population structure is essential to the premise of being able to identify associations with the PC4 mutational spectrum. In a normal GWAS, however, your analysis is hinging upon an assumption of the absence of population structure given that population structure could create a bunch of spurious associations with your trait, which is the reason for controlling for the first 20 genotype PCs. Spurious associations caused by population structure are already a serious problem for normal traits like height when they are strongly differentiated between subpopulations, even when investigators control for lots of genotype PCs. I unfortunately can't suggest a quick fix that would be able to test whether this analysis is trustworthy, but my sense is that proving this kind of GWAS is statistically valid would be enough work that it would need its own dedicated methods paper where you rigorously simulate an association between a mutation spectrum gradient and a causal allele, ascertain mutation spectra the way you are doing here, and test whether you are able to identify the correct locus using this pipeline. The discussion gives the caveat that the PARP11 association still needs to be replicated in additional data, but this seems like way too weak a caveat—it would be appropriate if this were a regular GWAS where you were directly measuring a regular phenotype in each individual. There is enough in the paper that it could still merit publication without this association test, so removing it and going on to write a methods paper proving its validity could be the best way to go.

Thanks for the comments. To check for the potential issue of population structure still confounding this analysis, after correcting for genotype PCs, we investigated whether there was evidence for inflation in the test statistic. The genomic inflation factor was though ~ 1 (1.006), suggesting there is no strong evidence of inflation of the test statistic due to uncorrected population structure (or other factors). This is broadly consistent with the fact that the samples separating in this mutation spectra analysis are samples that cluster closely on a standard genotype PCA. Following the revisions described in this document (for example excluding variants with a high derived allele frequency) the significance of this locus actually slightly increased ($P=5.3 \times 10^{-10}$). As the reviewer highlights, we had tried to be very careful and caveat this analysis, especially in the absence of replication. However, in accordance with the reviewer's suggestion we have now removed this analysis from the manuscript.

The quality filters that are imposed on the SNPs in this paper are quite lax: SNPs are only required to have 8x sequencing depth and less than 75% missing data. How were these quality thresholds chosen, and what is the evidence that they are good enough? I recognize that it isn't realistic to hold non-model-organism data to the same standards that would be realistic

for human data, but since there hasn't been any investigation of the possible effects of low data quality, I'm concerned that the paper's results might be substantially affected by bioinformatic error. Some of the early reports of mutation spectrum differences between human populations were later shown to be artifactual (Anderson-Trocme, et al. Mol Biol Evol 2020), so it is a real possibility that more serious artifacts could create some of the patterns in the current paper, especially when a lot is being made of the distribution of higher order PCs. One realistic quality check to implement would be to focus on the paper's main results (e.g. the TCC signature and PC4 gradient in cattle, the CpG transition deficit in pigs) and check whether these patterns still hold if the QC thresholds are strengthened to require say 20x depth and less than 20% missing data.

We don't believe the data is consistent with being strongly affected by sequencing coverage. Most notably, despite the range of coverages both within and across species, the samples still cluster by species and ancestry. Not by sequencing coverage, as would be expected if it was a major confounder. Demonstrating this, as suggested by the reviewer we specifically examined whether the PC4 gradient could potentially be confounded by sequencing depths by plotting PC4 values versus corresponding sample coverages. We have now added this plot as Supplementary Figure 8 and have also included it below:

As can be seen, African Indicine cattle have a positive PC4 value irrespective of their depth, with East Asian Indicine cattle showing a negative PC4, despite having similar coverages to many of the African Indicine samples. Likewise the primary Indicine split

is observed among samples of similar coverages. So coverage can not be the primary driver of this result. We have now added text highlighting this at line 270. This figure does also emphasise the difficulty if tried to restrict these non-human studies to only samples with depths above 20x as suggested. As can be seen, this would remove the majority of samples including largely all of the East Asian indicine samples.

The reproducible workflow that the authors have made would benefit a lot from the addition of some automated quality checks, and one could be a quantification of how much the k-SFS changes when the quality thresholds are varied. In particular, I noticed that there aren't any checks of the quality of the ancestral state identification being performed, which will depend a lot on how well the people running the pipeline have chosen the outgroup genome. Even in a best case scenario, the site frequency spectrum usually has a bit of a "smile" shape that indicates there are too many SNPs with frequencies close to 1—can the authors add a check of this 'smile' to the pipeline so that researchers will be aware if the ancestral state identification is not working very well? It is also best practice to discard SNPs above say a frequency of 90% since most of these SNPs will have misidentified ancestral alleles and will have the potential to confound the mutation spectrum.

Its true that SNPs above a derived allele frequency threshold have previously been excluded from similar analyses (a cutoff of 0.98 in the case of Harris and Pritchard). We did though have a concern about the potential issue of this kind of filter when looking across species and populations. If a MAF is calculated across all samples of a species, then populations with a larger sample size will contribute more to this filter. These then may lead to artefactual differences between the groups. For example if a variant has a real derived allele frequency of 20% in a larger population, but a low frequency in the other populations this variant is more likely to survive the filter than if it is common in a smaller population. One solution is to apply the MAF filter across all subpopulations. Though in cattle only 1% of variants have a derived allele frequency greater than 0.98 in all subpopulations. In contrast to 21% that have a derived allele frequency less than 0.02 in all subpopulations. So this would suggest few variants would be excluded by this filter. However, even this filter is not ideal as the shape of the distribution of allele frequencies is not the same across all populations, particularly the more admixed cattle groups, meaning, again, some populations will be the primary determinants of which variants will be filtered out. Likewise we were concerned that the use of smile plots for such QC may be less informative in species that have undergone bottleneck events, that may have led to derived alleles becoming fixed or nearly fixed.

Nevertheless, we also understand that some users would want the possibility to drop these variants, or to apply alternative thresholds of their choice. We modified nSPECTRa to accept a `--max_derivate_allele_freq` argument to the workflow (v1.1.1), defaulting to 0.98 as in Harris and Pritchard. It now also outputs "smile" plots as suggested and we have provided an example of such a plot in a response to reviewer 3 below. We have now rerun all of the analyses applying this default DAF cutoff, and accordingly updated all the results in the manuscript. This led to relatively few sites being dropped (approximately 1% in all species), and none of our main conclusions were effected. One issue that did arise is that for some human samples the median count of some SDMs was now 0, causing the median of ratio normalization to return infinite values. To avoid this problem, we added a small value to the K-mer-normalized counts. This small number is equal to 1 multiplied by 10^N , where N is the decimal notation of the smallest value in the matrix (e.g. if the smallest value is 3.5×10^{-9} , we add

1x10⁻⁹ to each K-mer-normalized value; lines 672-677). We checked the impact of this in the other species and found it had little impact on results. A second important issue we identified was with the jellyfish software used originally for K-mer counting, that can return different K-mer background counts in consecutive runs; this is now fixed by replacing jellyfish with a custom script.

The presence of excess TCC>TTC mutations in the indicine cattle is a cool result, but I think the paper overstates the strength of the evidence that this is the same as the mutational signature found in European and South Asian humans. The reason I say this is that the SHAP value plots don't seem to indicate an excess of the minor components of the European TCC pulse, including CCC>CTC, ACC>ATC, and TCT>TTT. There are at least two other examples of distinct mutational signatures that are also dominated by TCC>TTC but which have different minor components: the COSMIC UV light signature SBS18 and a signature found in the canine transmissible venereal tumor (Baez-Ortega, et al. Science 2019) where the TCC>TTC mutations have a biased pentanucleotide context distribution that is not seen in humans.

We agree that it is not possible to attribute the elevated rate of TCC>TTC mutations in human and cattle populations to the same underlying mechanisms, just that pulses for the same change are observed across the two species. We have now deleted the line in the discussion where it is suggested they may have the same underlying processes. Equally though it probably isn't possible to say that the CCC>CTC changes in European humans are driven solely by the same process as the TCC>TTC changes, as the former shows a markedly lower enrichment. In agreement with the reviewer's comments these spectra will be the result of various intersecting factors that together combine to influence their rates, some of which may or may not be shared across species.

Additional minor comments:

In line 59, it doesn't make sense to refer to SDMs as being caused by two 'independent' mutation events—if the first mutation increases the probability that the second mutation will occur, these are two statistically dependent events. It would be more accurate to say that an MNP is caused by a single concerted mutation while an SDM is caused by two mutations occurring at different times.

We have now changed this wording (line 59).

Lines 71-72 state that CpG>TpG mutations make up over a third of all observed mutations, and a third seems too high. I couldn't find where this figure came from when I skimmed the cited 2005 paper. If you look at Figure 2a of a more recent paper, Johnson, et al. Nature 2017, you see that CpG>TpG mutations are less than 20% of human germline de novo mutations. It would be better to specify what species you're talking about if you want to stand by the 1/3 figure, and whether you are talking about de novo mutations or polymorphisms

Many thanks for highlighting this. Apologies, this was a mistake. We have now changed line 72 as suggested.

I don't think lines 91-93 accurately represent Bergeron, et al.'s interpretation of their finding of elevated mutation rates in domesticated animals. As I understand it, Bergeron, et al. argued that population bottlenecks in domesticated animals may have relaxed selection on mutator

alleles, leading to increases in these species' mutation rates. This still counts as a change in these species' underlying mutational processes—without such a change, there's no reason that a historical population bottleneck could impact the rate or spectrum of de novo mutations. Some of the references are cited as preprints when published versions are available—the authors should update these as needed.

We have tweaked this sentence but we believe the above may not reflect the case laid out in Bergeron et al. The text below are direct quotes from their manuscript:

The higher mutation rates of domesticated animals are likely due to strong artificial selection for traits such as shorter generation times...Consequently, the higher yearly mutation rate observed in domesticated species is more likely to be explained by the lowering of reproductive age associated with domestication rather than by an inherent change to the mutational process.

So the authors are making the case that the unusually high mutation rates are a result of the shorter generation times induced by the process of domestication, rather than any inherent differences in mutational processes.

Thanks we have now updated the preprints.

Lines 100-102 say that “little work has been done to study mutation spectra differences outside of the major model organisms due to a lack of suitable large whole-genome sequenced cohorts and easy-to-use software tools.” Arguably the two most relevant papers to cite here would be Beichman, et al. 2023 and Bergeron, et al. 2023, which are not the two papers actually cited. (Bergeron, et al. were not able to study context-dependent mutation spectra but still looked at 1-mer mutation spectrum differences between vertebrate classes). I also think it comes across a bit strangely to say that “little work” has been done when in this case nonzero work has been done and it's a bit subjective to say whether the amount is little or big. As an alternative, the authors could either briefly describe what has been done or not done or else make some more precise claim about what they do that previous work has not done. In terms of why there has not been more work on mutation spectra in non-model organisms, I think the authors are missing what is perhaps the most important reason: non-model organism sequences tend to be of more variable bioinformatic quality than human and mouse sequences, so we have to worry more about reporting mutation spectrum differences that are really bioinformatic artifacts.

We have now cited the two paper mentioned here. Apologies, our intention was not to suggest no work outside of model organisms had been done but simply that more was needed. This is effectively also suggested in the Beichman et al. paper to which the reviewer refers where they say “*To better understand how genetics, environment, and age interact to shape the accumulation of mutations in the germline, more standardized mutation data from a variety of taxa will be needed.*” We have now added a note concerning the resource quality in the sentence at lines 95 to 97 as suggested.

Can the authors verify whether different filters based on SNP density might have been applied to the datasets they are using? Some bioinformatic pipelines discard clustered SNPs that might support MNPs or SDMs, so it is important to verify that this is not creating any artifactual SDM patterns.

If the authors are referring to the GATK --cluster-size hard filter then this was not applied to the locally processed datasets (cattle, water buffalo, African buffalo). The

VCF files from the GVM database (horse and pig) do not appear to have been called with the option, as documented here: https://ngdc.cnbc.ac.cn/gvm/analysis_standards. Similarly, the dog VCF from Plassais et al. was called using GATK standard parameters, as documented here: https://pmc.ncbi.nlm.nih.gov/articles/instance/6445083/bin/41467_2019_9373_MOESM1_ESM.pdf. Lastly, the 1000 genome project VCF was called using the method described in the supplementary methods of the publication that doesn't mention the aforementioned option (see https://static-content.springer.com/esm/art%3A10.1038%2Fnature15393/MediaObjects/41586_2015_BFnature15393_MOESM86_ESM.pdf).

The intro points out that Bergeron et al. did not test whether domesticated species have a different mutation spectrum from non-domesticated species, effectively implying that this current paper will fill that gap. However, this paper doesn't really address that question either, which would require pairing each domesticated species with a closely related non-domesticated species and testing whether there is some systematic difference between the domesticates and non-domesticates. It's fine that this paper doesn't tackle this question, but it seems misleading to imply that they are doing this by pointing out that Bergeron et al. did not do it, rather than just saying that the Bergeron results imply that domesticated species' mutation spectra might be interesting and you wanted to learn more about them.

In our introduction, all we say with respect to Bergeron et al. is “*However, the relatively few mutations identified when comparing parental and offspring genomes means that it is not possible to compare the spectra of mutations between species and among domesticated populations using such trio-based approaches.*” This is just saying that using trio-based approaches makes comparing mutation spectra between species/groups difficult. We have now clarified we mean mutation spectra in different kmer contexts and removed the mention of domesticated populations from the sentence (line 95).

Reviewer #3 (Remarks to the Author):

The manuscript entitled “The evolution and convergence of mutation spectra across mammals” authored by Talenti et al., describes a new workflow, nSPECTRA, to characterize mutation spectra. They demonstrate the power of the approach investigating the evolution of mutations in trinucleotides in domestic animal species and human. Their results are quite interesting, as they demonstrate differences between species, between and within populations, but also cases of convergent evolution between species.

I found the manuscript clear and very well written. It was a pleasure reading it. Here are my observations and suggestions:

1) The differences observed result from the different efficiency with which DNA repair mechanisms correct specific mutations. I wonder if some variation in these genes can be identified by association analyses. This should be possible particularly in the case of differences in mutation rates observed within a single population (e.g. N'Dama cattle). When happening in a same genetic background the effect is likely due to rather simple genetic variation that should be simpler to identify.

Thank you for the suggestion. We had tried this for the separation observed in Figure 4C (see previous Supplementary Figure 8), but didn't do a similar analysis for the separation in Figure 4B where the N'Dama are observed to separate because we were

concerned the sample size was too small. However, with the ever increasing number of cattle genomes being generated it would be very interesting to explore this further in the future when more data is available. Following the comments by reviewer 2, we have now removed this analysis altogether.

2) The divergence in mutation spectra within population is intriguing. My first interpretation was in favour of a trait not so important to estimate diversity (e.g. red and black Holstein differ in MC1R gene and coat colour but they are and will remain a single population), however there is a more profound effect, as with time DNA repair genes variation leads to profound genetic divergence. What appears to be a single event (SNV), starts a domino effect and with time induces MNPs and divergence. Can this be claimed as contributing to speciation in the long term?

Thanks for the very interesting comment. We think though in the absence of reproductive barriers mutation spectra changes alone would be unlikely lead to speciation. Effectively any pathway can be knocked out by any mutation spectra so differences in mutation spectra are unlikely to be major drivers. It would be the reproductive isolation that would be the primary driver of speciation.

3) Line 69. Selection, as well as drift, influences the distribution of allele frequencies in populations, but it's not clear how it influences the rate of mutation of some sequence. Can the authors clarify? I imagine they refer to the fact selection favors beneficial and purges deleterious mutation but this is not affecting "the rate by which certain sequences are prone to mutate". The mutational profile changes, but the rate of mutation should not.

Apologies for the misunderstanding, we were listing factors that influence the observed distribution of base changes in the genome. One factor is the rate at which sequences mutate, another factor is selection. Although these factors are not unrelated we were not meaning to imply one was influencing the other here. We have edited the text to clarify this (line 68).

4) line 120. The ability to infer ancestral allele on nSPECTRa would benefit from validation, e.g. using information on wild relatives of domestic species or ancient DNA data publicly available.

We chose to use CACTUS to determine ancestral alleles for multiple reasons, which importantly does incorporate information on wild relatives of domesticated species as suggested. While our workflow produces an ancestral genome matching the desired reference, the definition of the ancestral state of the individual positions is computed by CACTUS, with the workflow replacing the reference base with the respective ancestral state and is described in their manuscript here (<https://www.nature.com/articles/s41586-020-2871-y#Sec2>). We chose to use CACTUS specifically because it allows for reference-free alignments, and account for complex events in addition to small ones. Another notable reason that we chose CACTUS is it is the same approach now used by Ensembl to derive their ancestral genomes. Alternative choices, such as *est-sfs*, are limited to a specific topology (2 or 3 outgroups only), instead of a progressive improvement of the ancestral state propagating upwards in the tree. As a consequence, these two methods do not always match in the results (~91.5% of the 26M variants in the cattle dataset have matching ancestral state between

the two methods), and when faced with making a call, we decided to use CACTUS as an integrated solution capable of making alignments and defining the ancestral state.

The use of ancient sequencing data would likely come with extensive issues. The first issue is related to the sparsity and scattering of samples throughout history. For instance, the Erven et al. paper has samples ranging from the Neolithic to the middle ages;

<https://academic.oup.com/mbe/article/41/5/msae076/7658373?login=true#462477484>), which complicates the definition of the allele to be used as ancestral. The second, and arguably more relevant, issue for this study is that ancient samples tend to develop DNA damage in the form of deamination (C>T and G>A; <https://academic.oup.com/nar/article/29/23/4793/2359260>), in addition to contamination and challenges in achieving reasonably high coverage. Likewise suitable ancient genomes are not available for all species.

As requested by reviewer 2 we have now included a new output plot as part of the nSPECTRa workflow, showing the distribution of derived allele frequencies, under the assumption that if large numbers of ancestral alleles are being miscalled there will be a strong excess of variants with a high derived allele frequency. The plot below for the African buffalo variants, shows this is generally not the case supporting the idea that the ancestral allele calls are not substantially erroneous.

5) Figure 2b. In this figure I can't identify the high frequency of C>T transversions in in CGC trinucleotides (CGC \diamond TGC or CGT), as the deamination of methylated cytosines in CpG island is very frequent. Has this kind of mutation been left out of the analyses?

The data in this plot has been scaled so that each mutation type has a mean of 0 and a standard deviation of 1. This effectively puts each mutation type on the same scale and allows us to more easily compare the relative frequency between species. This does though mean that it isn't possible to compare relative levels between mutation types in this plot. We have now clarified this in the legend. The elevated rate of C>T

transversions in CpG contexts can though be seen in other figures, including Figure 2D.

6) lines 416-419. Have authors considered ancient contribution of wild relative species?

Thanks, we now added a comment in the discussion (lines 455-459).

7) lines 426-428. The hypothesis of two indicine domestications or at least two subgroups also results from mtDNA variation.

Thanks. We have added this to the discussion (line 442-443)

8) lines 459-460. Does this also justify the species-specific codon usage?

Thanks. We have added this (line 471-472).

Response to reviewers

Reviewer 2's remaining comment asked to address:

On the subject of randomly allocating each variant to one individual before performing a PCA, it's true that this might not have been done in the earliest mutation spectrum papers, but it's described in the methods of some more recent papers that rely heavily on mutation spectrum PCA (e.g. Goldberg and Harris 2022 <https://doi.org/10.1093/gbe/evab104> as well as Beichman, et al. 2023 <https://doi.org/10.1093/molbev/msad213>). In their response, the authors point out some legitimate disadvantages of this approach, including that changes the balance of rare and common alleles and it means that the mutation spectrum assigned to an individual depends on which other individuals are being jointly analyzed. In my view, these disadvantages are outweighed in the context of PCA analyses where the point is not to ascertain each population's mutation spectrum, but to test whether patterns of mutation spectrum similarity and divergence are sufficient to assign individuals to populations. The authors say that it isn't really a confounder if two individuals have similar mutational loads due to shared ancestry, but would they see it this way if two individuals in the dataset were siblings, in which case more than half of the mutations making up each sibling's mutation spectrum would be shared with the other sibling's mutation spectrum? Based on such a PCA, you would likely conclude that these two siblings had more similar mutation spectra than two unrelated individuals from the same population, even if the spectra of the siblings' non-shared mutations were as divergent as the spectra of two unrelated individuals. This sibling example is intentionally extreme, but each pair of unrelated individuals from the same species do often share many common genetic variants. For this reason, I'd suggest giving the approach a try or at least including a discussion section caveat saying that this random variant allocation wasn't done, meaning that shared variants might inflate some of the observed clustering patterns.

We have now added text to the discussion section regarding this point as follows (lines 403-412):

Furthermore, while early and foundational mutation-spectra studies generally counted variants in each carrier individually, one proposed approach when undertaking principal component analysis of mutation spectra is to randomly allocate each variant to a single carrier^{19,40}. This method can potentially help reduce inflated similarities due to shared ancestry or close relatedness. However, it also discards potentially important information concerning the overall mutational load of individuals, and the composition of the dataset (e.g. which samples are included) impacts the variants assigned to each sample and consequently their assayed mutation spectra. Hence, while random allocation may reduce over-clustering in certain scenarios, it also has the potential to introduce dataset-dependent biases.